# Multimodal Meta-learning of Implicit Neural Representations with Iterative Adaptation

## Abstract

Gradient-based meta-learning is gaining prominence in implicit neural representations (INRs) to accelerate convergence. However, existing approaches primarily concentrate on meta-learning *weight initialization* for *unimodal* signals. This focus falls short when data is scarce, as noisy gradients from a small set of observations can hinder convergence and trigger overfitting. Moreover, real-world data often stems from joint multimodal distributions, which share common or complementary information across modalities. This presents an opportunity to enhance convergence and performance, particularly when dealing with limited data. Unfortunately, existing methods do not fully exploit this potential due to their main focus on unimodal setups. In this work, we introduce a novel optimization-based meta-learning framework, Multimodal Iterative Adaptation (MIA), that addresses these limitations. MIA fosters continuous interaction among independent unimodal INR learners, enabling them to capture cross-modal relationships and refine their understanding of signals through iterative optimization steps. To achieve this goal, we introduce additional meta-learned modules, dubbed State Fusion Transformers (SFTs). Our SFTs are meta-learned to aggregate the states of the unimodal learners (*e.g.* parameters and gradients), capture their potential cross-modal interactions, and utilize this knowledge to provide enhanced weight updates and guidance to the unimodal learners. In experiments, we demonstrate that MIA significantly improves the modeling capabilities of unimodal meta-learners, achieving substantial enhancements in generalization and memorization performances over unimodal baselines across a variety of multimodal signals, ranging from 1D synthetic functions to real-world vision, climate, and audiovisual data.

## 1 Introduction

Implicit neural representations (INRs) are a class of neural networks that is designed to represent complex signals or data as coordinate-to-feature mapping functions. Since a variety of real-world data can be represented as functions, (*e.g.* an image is a function $I(x, y) = (r, g, b)$ mapping 2D coordinates to color intensity values), numerous efforts have been made to formulate data as functions and represent them with INRs in a wide variety of domains or modalities, including audios (Sitzmann et al., 2020), time series (Fons et al., 2022), images (Sitzmann et al., 2020), videos (Chen et al., 2021a), 3D geometries (Park et al., 2019) and scenes (Mildenhall et al., 2020). They have shown remarkable promise in applications such as compression (Dupont et al., 2022b; Kim et al., 2022b), generation (Skorokhodov et al., 2021; Yu et al., 2022), super-resolution (Chen et al., 2021b; 2022), and novel-view synthesis (Takikawa et al., 2021; Barron et al., 2023) scenarios.

However, fitting a neural network to such a function from randomly initialized weights has been found to be notoriously inefficient; for example, it requires hundreds or thousands of optimization steps to converge (Sitzmann et al., 2020; Mildenhall et al., 2020), and learned representations do not generalize well on novel input coordinates under the few-shot learning regimes (Tancik et al., 2021; Jain et al., 2021). To overcome such limitations, there have been increasing studies to apply gradient-based meta-learning techniques (Tancik et al., 2021; Dupont et al., 2022a) to facilitate the learning of INRs. The key idea is to pre-train a weight initialization that is close to the optimal weights for a wide range of signals, from which each online INR learner effectively adapts and generalizes to unseen signals given partial observations with a limited number of optimization steps.

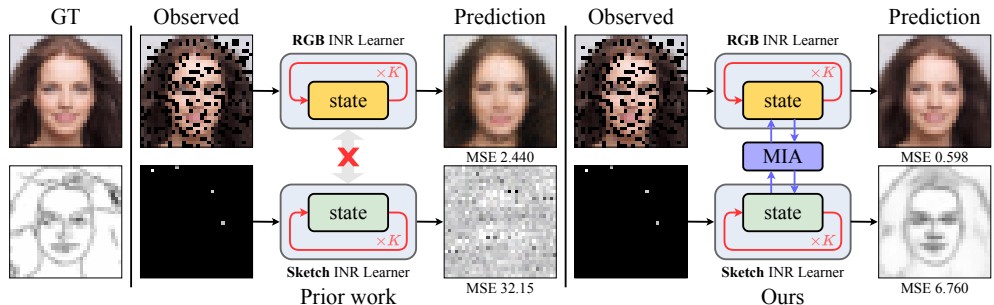

Figure 1: The existing methods primarily focus on learning meta-initialized weights on unimodal signals, only to adapt them separately in each modality without considering their potential interactions. In contrast, our proposed MIA facilitates the interaction among the independent learners and the capture of cross-modal relationships, enabling iterative refinement of their knowledge on the signals.

Despite the success, the majority of research predominantly focuses on meta-learning of *weight initialization* within *unimodal* signal distributions. This unimodal meta-initialization can pose significant limitations when data is scarce, since noises inherently arise in the gradient computation from a small set of observations (Zhang et al., 2019; Simon et al., 2020). This can be detrimental to the learning process, slowing down convergence and leading to overfitting. However, real-world signals often originate from joint multimodal distributions, where they frequently share common or complementary information with one another; for example, data collected jointly from multi-sensory systems such as medical devices, weather stations, or self-driving cars.

To this end, we introduce a new optimization-based meta-learning framework, featuring a novel Multimodal Iterative Adaptation (MIA) paradigm at its core. MIA facilitates continuous interaction across the independent unimodal INR learners by accessing to their learning progress or states (*e.g.* weights and gradients) and discovering their cross-modal relations through iterative optimization steps. This not only enables the effective capture of the intricate structures present in multimodal signal distributions, but also empowers more efficient navigation through the complex optimization landscapes, leading to enhanced modeling of multimodal signals with limited observations and a small number of optimization steps.

To achieve this goal, we augment MIA with additional meta-learned modules, coined State Fusion Transformers (SFTs). SFTs are meta-learned to achieve three objectives: (1) aggregating learning states from independent unimodal INR learners, (2) discovering rich unimodal and multimodal interactions within the states, and (3) leveraging this knowledge to enhance weight updates and guidance to individual INR learners throughout iterative adaptation procedures.

In experiments, we apply our technique to existing unimodal INR meta-learning frameworks such as Functa (Dupont et al., 2022a) and Composers (Kim et al., 2023), and evaluate them on a variety of multimodal signal regression scenarios, including modeling 1D synthetic functions (Kim et al., 2022a), ERA5 global climate data (Hersbach et al., 2019), 2D CelebA visual images (Xia et al., 2021), and Audiovisual-MNIST (AV-MNIST) data (Vielzeuf et al., 2018). The results consistently show that our multimodal adaptation strategy MIA significantly improves the modeling capability of the unimodal meta-learners, achieving at least $61.4\%$ and $81.6\%$ error reduction in generalization and memorization over the unimodal baselines, respectively.

## 2 PRELIMINARIES

**Implicit neural representations (INRs).** INRs are a class of neural networks, often parameterized as a stack of MLPs, that are designed to approximate a function $f : x \mapsto y$ mapping the coordinates $x$ to the features $y$. We interchangeably use data, signals, and functions throughout the paper. Given a set of coordinate-feature pairs $\mathcal{D} = \{(x^i, y^i)\}_{i=1}^P$ of a function, an INR $f_{\text{INR}}(\cdot; \theta)$ with randomly initialized weights $\theta$ is optimized to minimize a loss function as below:

$$\mathcal{L}(\mathcal{D}; \theta) = \frac{1}{P} \sum_{i=1}^{P} ||f_{\text{INR}}(x^i; \theta) - y^i||_2^2. \tag{1}$$

Despite the recent advances, optimizing individual INRs from randomly initialized weights $\theta$ are notoriously inefficient (Sitzmann et al., 2020; Mildenhall et al., 2020) and they often do not generalize

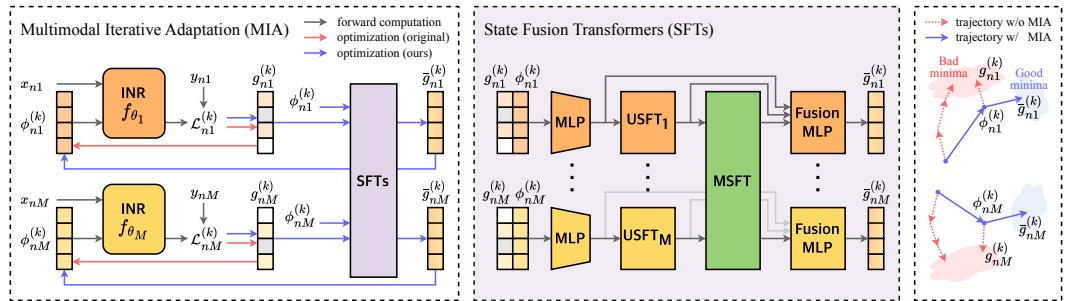

Figure 2: An illustration on our proposed Multimodal Iterative Adaptation (MIA). Left: At each step $k$, each INR learner ($f_{\theta_m}$) calculates the loss ($\mathcal{L}_{nm}^{(k)}$) given the parameters ($\phi_{nm}^{(k)}$) and data ($x_{nm}, y_{nm}$) during the forward pass (black arrow), followed by computing gradients ($g_{nm}^{(k)}$) to update the parameters. Before updating the parameters with the gradients (red arrow), we enhance them via State Fusion Transformers (SFTs) to better guide the learners (blue arrow). Middle: SFTs aggregate the states ($\{\phi_{nm}^{(k)}, g_{nm}^{(k)}\}_m$), capture the unimodal and cross-modal interactions respectively via USFTs and MSFTs, and fuse this knowledge to compute enhanced updates ($\{\bar{g}_{nm}^{(k)}\}_m$) via Fusion MLPs. Right: This guidance accelerates convergence of the learners to better solutions (See Appendix F.3).

well on few-shot regimes (Tancik et al., 2021; Jain et al., 2021), *i.e.* when $P$ is small. To address these problems, gradient-based meta-learning techniques have been proposed.

**Meta-Learning Approach**. Two notable examples of such meta-learning frameworks are Functa (Dupont et al., 2022b;a; Bauer et al., 2023) and Instance Pattern Composers (Kim et al., 2023) (*i.e.* Composers for brevity). They have shown promise in modeling various types of complex signals thanks to its modality-agnostic and scalable design. These methods build upon CAVIA (Zintgraf et al., 2018) and implement two key ideas: (1) They meta-learn INR parameters $\theta$ that capture data-agnostic knowledge, providing strong priors that enable faster convergence when modeling each signal. (2) They introduce additional context parameters $\phi \in \mathbb{R}^{S \times D}$, a set of $D$-dimensional features[1] that are adapted to each signal. These features encapsulate data-specific variations and conditions of the meta-learned INRs $f_{\text{INR}}(\cdot; \theta)$ via modulation schemes (Perez et al., 2018; Schwarz et al., 2023).

Formally, given a dataset $\mathcal{D} = \{\mathcal{D}_n\}_{n=1}^N$ of $N$ functions, the objective in Eq (1), which fits an INR to a function $\mathcal{D}_n = \{(x_n^i, y_n^i)\}_{i=1}^{P_n}$, is extended to accommodate both of the parameters ($\theta, \phi$) as below:

$$\mathcal{L}(\mathcal{D}_n; \theta, \phi_n) = \frac{1}{P_n} \sum_{i=1}^{P_n} ||f_{\text{INR}}(x_n^i; \theta, \phi_n) - y_n^i||_2^2. \tag{2}$$

Then, INR weights $\theta$ and context parameters $\phi$ are optimized with the meta-objective as follows:

$$\theta, \phi = \operatorname*{argmin}_{\theta, \phi} \mathbb{E}_n \left[ \mathcal{L}(\mathcal{D}_n; \theta, \phi_n^{(K)}) \right], \text{ where} \tag{3}$$

$$\phi_n^{(k)} = \phi_n^{(k-1)} - \alpha \cdot \nabla_{\phi_n^{(k-1)}} \mathcal{L}(\mathcal{D}_n; \theta, \phi_n^{(k-1)}), \text{ for } k = 1, \dots, K. \tag{4}$$

Here, $\alpha$ is a learning rate and $\phi_n^{(0)} = \phi$. Intuitively, the objective can be cast as a bi-level optimization problem: (1) In the inner loop (Eq. 4), each online INR learner is first initialized with weights $\theta$ and $\phi^{(0)}$ and then adapted to each function $\mathcal{D}_n$ through $\phi^{(k)}$ ($k = 1, \dots, K$). (2) In the outer loop (Eq. 3), the meta learner updates the INR weights $\theta$ and context parameters $\phi$ so that each fitted INR $f_{\text{INR}}(\cdot; \theta, \phi_n^{(K)})$ represents the function $\mathcal{D}_n$ well within $K$ steps.

While these methods are competitive baselines for modeling complex signals across various modalities, their applicability is mainly demonstrated in memorizing unimodal signals when sufficient data is available due the inability to generalize well with limited data and consider cross-modal interactions.

## 3 APPROACH

We delve into our novel gradient-based meta-learning framework for INRs, which effectively captures and leverages cross-modal interactions. We first extend the previous unimodal frameworks to

---

[1]In Functa, it is parameterized as a grid of local features while Composers adopt non-local ones.

multimodal setups in a straightforward way. We then present MIA, a novel adaptation scheme that allows independent unimodal learners to exchange their current learning progress and acquired knowledge about the signals through attention mechanism. Figure 2 overviews our framework.

We consider a dataset $\mathcal{D} = \{\mathcal{D}_n\}_{n=1}^N$ with $N$ pairs of multimodal signals. Each pair $\mathcal{D}_n = \{\mathcal{D}_{nm}\}_{m=1}^M$ is composed of signals from $M$ different modalities. Each signal $\mathcal{D}_{nm} = \{(x_{nm}^i, y_{nm}^i)\}_{i=1}^{P_{nm}}$ contains $P_{nm}$ coordinate-feature points which could vary with signals.

**A Naive Framework for Multimodal Signals.** We begin with a simple baseline that combines per-modality meta-learners together while treating each of them separately. This framework consists of a set of independent per-modality INR parameters $\theta = \{\theta_m\}_{m=1}^M$ and context parameters $\phi = \{\phi_m\}_{m=1}^M$. Then, the loss function for a signal $\mathcal{D}_{nm} = \{(x_{nm}^i, y_{nm}^i)\}_{i=1}^{P_{nm}}$ is formulated as

$$\mathcal{L}(\mathcal{D}_{nm}; \theta_m, \phi_{nm}) = \frac{1}{P_{nm}} \sum_{i=1}^{P_{nm}} ||\hat{y}_{nm}^i - y_{nm}^i||_2^2, \quad \text{where } \hat{y}_{mn}^i = f_{\text{INR}}(x_{mn}^i; \theta_m, \phi_{nm}). \quad (5)$$

Next, we modify the meta-objective to promote both memorization and generalization performances during the optimization. For this, we split each data into support $\mathcal{D}_{nm}^{\text{train}}$ and query $\mathcal{D}_{nm}^{\text{val}}$ sets, and use the following gradient-based algorithm to meta-learn the parameters for each modality:

$$\theta_m, \phi_m = \underset{\theta_m, \phi_m}{\operatorname{argmin}} \, \mathbb{E}_n \left[ \mathcal{L}(\mathcal{D}_{nm}^{\text{val}}; \theta_m, \phi_{nm}^{(K)}) \right], \text{ where} \quad (6)$$

$$\phi_{nm}^{(k+1)} = \phi_{nm}^{(k)} - \alpha_m \cdot g_{nm}^{(k)}, \quad g_{nm}^{(k)} = \nabla_{\phi_{nm}^{(k)}} \mathcal{L}(\mathcal{D}_{nm}^{\text{train}}; \theta_m, \phi_{nm}^{(k)}), \quad (7)$$

for $k = 0, \dots, K-1$ with $\phi_{nm}^{(0)} = \phi_m$. $\alpha_m$ is a learning rate for $\phi_{nm}^{(k)}$ of each modality.

**Multimodal Iterative Adaptation (MIA).** In the above parametrizations, each unimodal learner is adapted separately. Thus, we propose to adapt them jointly via our MIA, which facilitates their iterative interaction and promotes seamleass refinement of their understanding of the signals.

The core component in MIA is State Fusion Transformers (SFTs), additional meta-learned modules that aim at offering improved guidance to INR learners. This is achieved via a three-step process: (1) aggregating the states of the unimodal learners, (2) capturing both unimodal and cross-modal interactions to promote the knowledge exchange across them, (3) and utilizing this knowledge to improve the weight updates of the learners. The SFTs consist of Unimodal State Fusion Transformers (USFTs), Multimodal State Fusion Transformers (MSFTs), and Fusion MLPs.

For each modality, we first compute state representations $z_{nm}^{(k)} \in \mathbb{R}^{S_m \times D_z}$ of unimodal learners. We concatenate the context parameters $\phi_{nm}^{(k)} \in \mathbb{R}^{S_m \times D_\phi}$ and gradients $g_{nm}^{(k)} \in \mathbb{R}^{S_m \times D_\phi}$ along the feature dimension, followed by a projection MLPs. Then, the representations are fed into USFTs with $L_1$ transformer blocks and per-modality parameters $\xi_m$:

$$\hat{z}_{nm}^{(k)} = \text{USFT}_m(z_{nm}^{(k)}; \xi_m), \text{ for } m = 1, \dots, M, \quad (8)$$

where $\hat{z}_{nm}^{(k)}$ are updated state representations by USFTs, which dedicate to modeling dependencies among a set of states within a modality. Next, we combine these per-modality representations into a sequence $\hat{z}_n^{(k)} = \{\hat{z}_{nm}^{(k)}\}_{m=1}^M$ and input to the MSFTs with $L_2$ transformer blocks and shared parameters $\xi_s$. This results in more comprehensive multimodal state representations $\tilde{z}_n^{(k)}$ considering potential cross-modal interactions.

$$\tilde{z}_n^{(k)} = \text{MSFT}(\hat{z}_n^{(k)}; \xi_s), \text{ where } \hat{z}_n^{(k)} = \{\hat{z}_{nm}^{(k)}\}_{m=1}^M. \quad (9)$$

The final step is to integrate them all and generate enhanced weight updates for the learners. We concatenate the computed per-modality state representations $z_{mn}^{(k)}, \hat{z}_{mn}^{(k)}, \tilde{z}_{mn}^{(k)} \in \mathbb{R}^{S \times D_z}$ along the feature dimension, followed by processing each of the features with Fusion MLPs:

$$\bar{g}_{mn}^{(k)} = \text{FusionMLP}(\bar{z}_{mn}^{(k)}; \xi_m), \text{ where } \bar{z}_{mn}^{(k)} = [z_{mn}^{(k)} \| \hat{z}_{mn}^{(k)} \| \tilde{z}_{mn}^{(k)}]. \quad (10)$$

Here, $\bar{g}_n^{(k)} = \{\bar{g}_{mn}^{(k)}\}_{m=1}^M$ is the generated weight updates for the unimodal learners. They are used to adapt $\phi_{nm}^{(k)}$ for each inner step $k = 0, \dots, K-1$, while the parameters of SFTs $\xi = \xi_s \cup \{\xi_m\}_{m=1}^M$ along with those of INRs $\theta$ and $\phi$ are meta-learned in the outer-optimization stage:

$$\theta, \phi, \xi = \underset{\theta, \phi, \xi}{\operatorname{argmin}} \, \mathbb{E}_{n,m} \left[ \mathcal{L}(\mathcal{D}_{nm}^{\text{val}}; \theta_m, \phi_{nm}^{(K)}) \right], \quad \text{where } \phi_{nm}^{(k+1)} = \phi_{nm}^{(k)} - \bar{g}_{nm}^{(k)}. \quad (11)$$

Note that utilizing attention mechanisms for multimodal fusion and adopting meta-learned modules for enhanced weight updates are actively studied concepts in their respective domains of multimodal learning (Bachmann et al., 2022) and learning to optimize (Baik et al., 2023). To the best of our knowledge, however, none of the methods have explored the marriage of both, which holds great potential in effective navigation of the multimodal signal distributions and their optimization landscapes.

## 4 RELATED WORK

Training INRs from scratch has proven to be an expensive endeavor, despite their notable success in various applications. Meta-learning has emerged as a promising solution to address this challenge.

**Encoder-based INR Learning.** This class of methods utilizes encoder networks for predicting the parameters of INRs. While the majority of these methods primarily focus on modeling unimodal signal distributions, Multitask Neural Processes (MTNPs) (Kim et al., 2022a) aim to approximate multimodal signal distributions by incorporating multimodal feed-forward encoder networks with specialized attention and pooling mechanisms tailored for this setup. Nevertheless, this class of methods often suffers from underfitting issues (Garnelo et al., 2018), especially when dealing with complex signals. Careful parametrization of the encoder and INR architectures is found to be critical to alleviate such problems (Kim et al., 2019; Guo et al., 2023; Mehta et al., 2021; Kim et al., 2023).

**Optimization-based Meta-learning.** Drawing inspiration from well-established model-agnostic meta-learning algorithms like MAML (Finn et al., 2017) and CAVIA (Zintgraf et al., 2018), these methods primarily concentrate on learning meta-initialization to enhance the efficiency of computation and sample usage (Tancik et al., 2021; Bauer et al., 2023). While effective in many cases, the sole reliance of such meta-initialized weights often encounters challenges when facing poorly conditioned optimization landscapes (Vicol et al., 2021) or dealing with limited data due to high variances and noises in gradients (Simon et al., 2020). Moreover, existing methods mostly focus on unimodal setups, overlooking potential cross-modal interactions for enhanced convergence and performance.

**Learning to optimize (L2O).** Our method is related to the work in L2O, a complementary meta-learning approach that emphasizes enhancing weight optimization algorithms, rather than on pre-training weight initializations. The common objective is to empower an optimizer to meta-learn underlying structures of data and accelerate convergence through gradient preconditioning (Kang et al., 2023) and learning rate scheduling (Baik et al., 2023). For this, L2O methods have explored the utilization of various state features of learners, including parameters, gradients, and their higher-order statistics and relations (Gärtner et al., 2023), similar to ours. To the best of our knowledge, however, none of them have explored the potential of utilizing their cross-modal structures as ours.

## 5 EXPERIMENTS

We demonstrate that our proposed multimodal iterative adaptation strategy, MIA, indeed improves convergence and generalization of state-of-the-art meta-learning INR models. More comprehensive qualitative results can be found in Appendix A.

**Baselines.** We choose Functa (Dupont et al., 2022a) and Composers (Kim et al., 2023) as our base INR models upon which our framework and other baselines build. They are simply referred to as CAVIA in the following sections. As baselines, we consider recent L2O approaches that can be easily integrated into CAVIA, including MetaSGD (Li et al., 2017), GAP (Kang et al., 2023), and ALFA (Baik et al., 2023). They are partly similar to ours since they aim to meta-learn a set of parameters to accelerate the convergence by gradient preconditioning. However, unlike ours, these methods do not consider potential cross-modal interactions between the learners. Also, we include MTNPs (Kim et al., 2022a) as our baseline, which is based on Neural Processes (NPs) (Garnelo et al., 2018) and designed to approximate multimodal signals. Unlike ours, this method replaces the iterative optimization process with multimodal encoder networks with specialized attention and pooling mechanisms. Lastly, to isolate the impact of specific model design or parameterization, we introduce an additional encoder-based meta-learning baseline. This method employs the encoder parameterized with the same transformer architecture backbone as our SFTs yet uses it to directly predict the parameters of INRs, instead of adapting them via optimization. We refer to this baseline as Encoder. For fair comparisons, we use the same INR architecture backbones for all the methods. We set $K = 3$ for the optimization-based methods. Please refer to Appendix D.2 for the details.

Table 1: Quantitative comparisons on the multimodal 1D synthetic functions. We report the normalized MSEs ($\times 10^{-2}$) computed over distinct ranges of sampling ratios, averaged over 5 random seeds.

| Modality | | Sine | | | Gaussian | | | Tanh | | | ReLU | | |
|---|---|---|---|---|---|---|---|---|---|---|---|---|---|
| Range | $R^{\min}$ | 0.01 | 0.02 | 0.05 | 0.01 | 0.02 | 0.05 | 0.01 | 0.02 | 0.05 | 0.01 | 0.02 | 0.05 |
| | $R^{\max}$ | 0.02 | 0.05 | 0.10 | 0.02 | 0.05 | 0.10 | 0.02 | 0.05 | 0.10 | 0.02 | 0.05 | 0.10 |
| Functa | | 45.45 | 16.58 | 3.316 | 19.83 | 4.603 | 1.043 | 23.02 | 3.788 | 0.587 | 84.49 | 13.54 | 2.992 |
| w/ MetaSGD | | 44.69 | 16.46 | 3.326 | 19.75 | 4.334 | 1.055 | 26.16 | 4.528 | 0.718 | 69.86 | 10.45 | 2.044 |
| w/ GAP | | 44.76 | 16.49 | 3.297 | 19.54 | 4.518 | 1.052 | 23.46 | 3.799 | 0.555 | 63.17 | 10.12 | 1.862 |
| w/ ALFA | | 48.09 | 18.67 | 5.160 | 19.09 | 4.910 | 1.580 | 22.45 | 4.200 | 0.872 | 53.83 | 8.303 | 1.536 |
| w/ MTNPs | | 13.34 | 4.718 | 1.871 | 2.019 | 1.513 | 1.285 | 4.678 | 0.794 | 0.340 | 17.75 | 1.315 | 0.398 |
| w/ Encoder | | 7.746 | 3.157 | 1.409 | 1.287 | 0.972 | 0.838 | 1.305 | **0.349** | **0.114** | 5.980 | **0.720** | 0.122 |
| w/ **MIA** | | **6.386** | **2.058** | **0.547** | **1.057** | **0.571** | **0.281** | **1.285** | 0.378 | 0.131 | **5.069** | 1.012 | **0.115** |
| Composers | | 37.90 | 17.40 | 5.539 | 6.916 | 3.495 | 1.584 | 14.90 | 3.974 | 0.975 | 50.02 | 11.75 | 3.426 |
| w/ MetaSGD | | 38.26 | 17.28 | 5.427 | 6.887 | 3.221 | 1.388 | 14.99 | 3.938 | 0.981 | 51.97 | 11.65 | 3.530 |
| w/ GAP | | 37.53 | 17.52 | 5.397 | 6.630 | 3.409 | 1.526 | 14.40 | 3.828 | 0.978 | 50.90 | 10.85 | 3.128 |
| w/ ALFA | | 36.53 | 14.87 | 4.115 | 5.650 | 2.770 | 1.154 | 14.18 | 3.426 | 0.799 | 42.96 | 6.814 | 1.481 |
| w/ MTNPs | | 16.62 | 4.859 | 0.766 | 2.256 | 1.252 | 0.708 | 4.670 | 0.743 | 0.121 | 11.47 | 0.897 | **0.114** |
| w/ Encoder | | 8.385 | 3.878 | 1.939 | 1.332 | 1.097 | 0.906 | 1.348 | 0.371 | **0.111** | 5.745 | **0.719** | 0.134 |
| w/ **MIA** | | **5.564** | **1.844** | **0.627** | **0.975** | **0.528** | **0.237** | **1.257** | **0.343** | 0.128 | **4.715** | 0.943 | 0.156 |

**Datasets & Metrics.** We demonstrate the versatility of our method in four datasets: (1) 1D synthetic functions (Kim et al., 2022a), (2) 2D CelebA images (Xia et al., 2021), (3) ERA5 global climate dataset (Hersbach et al., 2019), and (4) Audiovisual-MNIST (AV-MNIST) dataset (Vielzeuf et al., 2018). To construct the support set $\mathcal{D}_{nm}^{\text{train}}$, we sample $P_{nm} \times R_{nm}$ random coordinate-feature pairs from the full set $\mathcal{D}_{nm} = \{(x_{nm}^i, y_{nm}^i)\}_{i=1}^{P_{nm}}$, where the sampling ratio $R_{nm} \in [R_m^{\min}, R_m^{\max}]$ is drawn independently for each signal. We use the full set as the query set $\mathcal{D}_{nm}^{\text{val}}$ during the outer loop evaluation to promote both memorization and generalization. Please find Appendix D.1 for more details. For comparisons, we report mean squared errors (MSEs) over distinct ranges of sampling ratios to investigate the models' ability to generalize to signals under different support set sizes.

## 5.1 MULTIMODAL 1D SYNTHETIC FUNCTIONS

**Setups.** Following prior works (Finn et al., 2017; Guo et al., 2020; Kim et al., 2022a), we begin with a simple yet controlled 1D synthetic function regression setting. Specifically, we first define a canonical form of parametric multimodal functions as below:

$$y_{nm} = a_n \cdot \text{Act}_m(b_n \cdot x_{nm} + c_n) + d_n, \text{ where } \text{Act}_m \in \{\text{Sine}, \text{Gaussian}, \text{Tanh}, \text{ReLU}\}. \quad (12)$$

Each function $\mathcal{D}_{nm}$ is instantiated with the shared parameters $(a_n, b_n, c_n, d_n)$ and modality-specific non-linear activation functions $\text{Act}_m(\cdot)$. We follow Kim et al. (2022a) to construct the dataset: We define a uniform grid of $P_{nm} = 200$ coordinates within a range of $x \in [-5, 5]$. We sample $N = 1000$ function parameters $(a_n, b_n, c_n, d_n)$ shared across modalities and add per-modality Gaussian noises $\epsilon_{nm} \sim \mathcal{N}(0, 0.02)$ to control the cross-modal correlations. We use $[R_m^{\min}, R^{\max}] = [0.01, 0.1]$.

**Results.** We present the quantitative results in Table 1, where we report normalized MSEs by the scale parameter $a_{nm}$ per function, $\text{MSE} = \frac{1}{N} \sum_{n=1}^{N} \frac{1}{a_{nm}^2} \|\hat{y}_{nm} - y_{nm}\|_2^2$, following Kim et al. (2022a). The methods with no ability to handle multimodal signal jointly (CAVIA, MetaSGD, GAP, ALFA) fail to approximate the functions, showing high MSEs in all ranges of sampling ratios. In contrast, the multimodal methods (MTNP, Encoder, and ours) are able to reconstruct each signal more precisely, even with extremely small number of support sets (*e.g.* $R < 0.02$). Moreover, while MTNPs and Encoder show strong fitting performances on smooth and low-curvature signals (*i.e.* Tanh and ReLU) when data is sufficient, our method achieves the best performances overall among the multimodal methods, verifying its effectiveness in utilizing cross-modal interactions to enhance generalization.

## 5.2 MULTIMODAL 2D CELEBA DATASET

**Setups.** We conduct experiments on more complex 2D image function regression settings (Tancik et al., 2021; Kim et al., 2022a). We consider three different visual modalities of 2D facial data based on CelebA (Liu et al., 2015; Xia et al., 2021), namely RGB images (Karras et al., 2018), surface normal maps [2], and sketches (Xia et al., 2021). Following (Kim et al., 2022a), we resize all data to $32 \times 32$ resolution. We interpret them as functions mapping 2D coordinates $x \in [-1, 1]^2$ to output

---

[2]We use a pre-trained model (Eftekhar et al., 2021) to obtain surface normals on CelebA-HQ RGB images.

Table 2: Results on the multimodal 2D CelebA image function regression. We report the MSEs ($\times 10^{-3}$) computed over distinct ranges of sampling ratios, averaged over 5 different seeds.

| Modality | | RGBs | | | | Normals | | | | Sketches | | | |
|---|---|---|---|---|---|---|---|---|---|---|---|---|---|
| Range | $R^{\min}$ $R^{\max}$ | 0.00 0.25 | 0.25 0.50 | 0.50 0.75 | 0.75 1.00 | 0.00 0.25 | 0.25 0.50 | 0.50 0.75 | 0.75 1.00 | 0.00 0.25 | 0.25 0.50 | 0.50 0.75 | 0.75 1.00 |
| Functa | | 13.32 | 4.008 | 2.900 | 2.408 | 5.079 | 2.401 | 2.028 | 1.859 | 13.57 | 6.966 | 5.368 | 4.756 |
| w/ MetaSGD | | 13.02 | 3.830 | 2.685 | 2.182 | 4.923 | 2.268 | 1.864 | 1.682 | 12.72 | 6.278 | 4.532 | 3.839 |
| w/ GAP | | 12.84 | 3.726 | 2.543 | 2.024 | 4.805 | 2.184 | 1.762 | 1.570 | 12.43 | 6.023 | 4.166 | 3.407 |
| w/ ALFA | | 11.83 | 3.257 | 1.957 | 1.362 | 4.115 | 1.806 | 1.285 | 1.014 | 10.80 | 4.801 | 2.463 | 1.283 |
| w/ MTNPs | | 9.871 | 4.807 | 4.105 | 3.644 | 3.983 | 2.552 | 2.339 | 2.221 | 9.680 | 6.568 | 5.395 | 4.819 |
| w/ Encoder | | 7.810 | 3.266 | 2.486 | 2.042 | 3.463 | 1.934 | 1.597 | 1.412 | 8.645 | 5.222 | 3.658 | 2.800 |
| w/ **MIA** | | **6.946** | **2.563** | **1.627** | **1.135** | **2.979** | **1.530** | **1.118** | **0.869** | **7.667** | **4.042** | **2.142** | **1.011** |
| Composers | | 22.41 | 12.41 | 11.09 | 10.24 | 8.613 | 6.583 | 6.415 | 6.292 | 19.17 | 15.73 | 14.88 | 14.63 |
| w/ MetaSGD | | 20.11 | 10.25 | 9.000 | 8.268 | 8.218 | 5.979 | 5.753 | 5.601 | 18.95 | 15.43 | 14.49 | 14.20 |
| w/ GAP | | 20.07 | 10.05 | 8.785 | 8.039 | 8.149 | 5.847 | 5.616 | 5.461 | 18.58 | 15.24 | 14.39 | 14.15 |
| w/ ALFA | | 15.12 | 4.887 | 3.376 | 2.681 | 5.444 | 2.399 | 1.773 | 1.469 | 12.57 | 5.984 | 3.416 | 2.124 |
| w/ MTNPs | | 9.902 | 4.957 | 4.269 | 3.813 | 4.184 | 2.747 | 2.545 | 2.437 | 9.791 | 6.425 | 5.163 | 4.544 |
| w/ Encoder | | 10.21 | 4.278 | 3.205 | 2.599 | 4.106 | 2.287 | 1.889 | 1.665 | 9.904 | 5.602 | 3.697 | 2.644 |
| w/ **MIA** | | **9.764** | **3.418** | **1.913** | **1.017** | **3.749** | **1.763** | **1.062** | **0.526** | **9.505** | **4.708** | **2.336** | **0.855** |

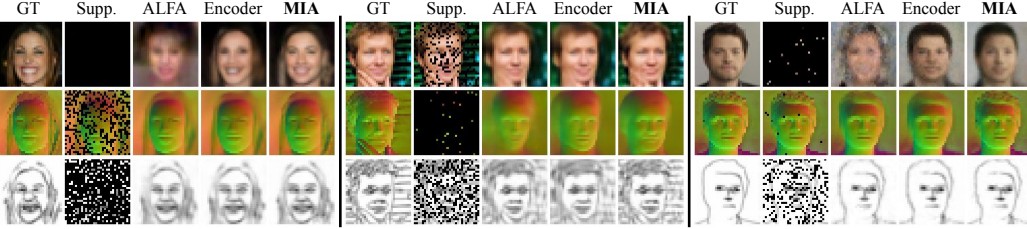

Figure 3: Qualitative comparisons on CelebA. Compared to the baselines, our MIA better generalizes in the extremely low-shot setting (first pane) and renders more rich high-frequency details or nuances in RGB (second pane), Sketch (second pane), and Normal (third pane) images when data is sufficient.

features $y \in \mathbb{R}^d$, where $d$ varies with the modality. We use 90% of the entire dataset for meta-training and the rest for the meta-testing, and set $[R_m^{\min}, R_m^{\max}] = [0.001, 1.000]$ for all the modalities.

**Results.** Table 2 summarizes the quantitative results. Similar to the synthetic scenario, the multimodal baselines with the encoders (MTNPs, Encoder) show better generalization over the unimodal baselines (CAVIA, MetaSGD, GAP, ALFA) under low-shot settings ($R \leq 0.25$). However, given sufficient number of support sets ($R \geq 0.50$), the multimodal encoder baselines perform worse than the unimodal counterpart ALFA. This could be attributed as underfitting phenomenon (Kim et al., 2019; Guo et al., 2023) suffered by the encoder-based meta-learning approaches. Unlike the baselines, our proposed method outperforms the baselines in all the ranges of support set sizes. In Figure 3, we confirm our method produces the most pleasing qualitative results, showing strong generalization with limited data while preserving high-frequency details and noises when sufficient data is available.

## 5.3 Multimodal Climate Data

**Setups.** ERA5 (Hersbach et al., 2019) is a global climate dataset that provides hourly estimates on a wide range of atmospheric variables measured globally over a grid of equally spaced latitudes and longitudes. Out of all the available variables, we consider the measurements on temperature, pressure, and humidity. Following Dupont et al. (2022c), we use a grid resolution of $46 \times 90$, and normalize the coordinates as $x \in [-1, 1]^2$ and features as $y \in [0, 1]$. We use 80% of the dataset for meta-training while using the rest for meta-testing, and set $[R_m^{\min}, R_m^{\max}] = [0.001, 1.000]$ for all the modalities.

**Results.** As shown in Table 3, ALFA shows decent performances out of all the unimodal baselines, even outperforming both multimodal baselines in the absence and abundance of data (*e.g.* when parameterized with Functa and in Pressure modality). This could be explained by the property of atmospheric variables that are relatively stable over time and the regions, where the fast convergence could be the key to excel in this setup. Finally, our method outperforms all the baselines, validating its effectiveness in modeling real-world multimodal climate data (results in Fig. 4 support this claim).

## 5.4 Multimodal Audio-Visual AVMNIST Dataset

**Setups.** Unlike previous setups, modeling audiovisual signals presents unique challenges arising from the heterogeneity in coordinate systems and lack of explicit spatiotemporal alignments between

Table 3: Results on ERA5 dataset in MSEs ($\times 10^{-5}$) across different sampling ratios.

| Modality | | Temperature | | | Pressure | | | Humidity | | |
|---|---|---|---|---|---|---|---|---|---|---|
| Range | $R^{\min}$ / $R^{\max}$ | 0.00 / 0.25 | 0.25 / 0.50 | 0.50 / 1.00 | 0.00 / 0.25 | 0.25 / 0.50 | 0.50 / 1.00 | 0.00 / 0.25 | 0.25 / 0.50 | 0.50 / 1.00 |
| Functa | | 6.83 | 3.53 | 3.38 | 1.42 | 0.98 | 0.95 | 33.7 | 23.4 | 21.8 |
| w/ MetaSGD | | 6.10 | 3.13 | 2.97 | 1.41 | 0.95 | 0.92 | 31.4 | 20.5 | 18.4 |
| w/ GAP | | 6.57 | 2.73 | 2.47 | 1.45 | 0.66 | 0.58 | 32.3 | 22.0 | 19.7 |
| w/ ALFA | | 4.07 | 1.56 | 1.26 | 0.62 | 0.19 | 0.15 | 27.7 | 15.4 | 11.4 |
| w/ MTNPs | | 4.72 | 3.86 | 3.75 | 1.45 | 1.40 | 1.38 | 29.8 | 27.5 | 27.0 |
| w/ Encoder | | 2.59 | 1.55 | 1.41 | 0.65 | 0.35 | 0.32 | 17.3 | 14.4 | 13.5 |
| w/ **MIA** | | **2.02** | **1.01** | **0.72** | **0.42** | **0.15** | **0.11** | **15.8** | **11.4** | **8.22** |
| Composers | | 7.91 | 7.26 | 7.19 | 2.29 | 2.21 | 2.20 | 48.9 | 48.3 | 48.1 |
| w/ MetaSGD | | 8.00 | 7.26 | 7.19 | 2.21 | 2.16 | 2.15 | 39.9 | 38.6 | 38.4 |
| w/ GAP | | 8.30 | 7.48 | 7.40 | 2.33 | 2.28 | 2.27 | 38.1 | 35.2 | 34.7 |
| w/ ALFA | | 7.03 | 6.17 | 6.06 | 1.91 | 1.81 | 1.79 | 37.4 | 33.6 | 33.0 |
| w/ MTNPs | | 4.49 | 3.62 | 3.53 | 1.39 | 1.33 | 1.31 | 29.6 | 26.9 | 26.3 |
| w/ Encoder | | 4.28 | 3.18 | 3.06 | 1.34 | 1.11 | 1.08 | 24.8 | 21.6 | 21.0 |
| w/ **MIA** | | **3.83** | **2.46** | **1.34** | **1.10** | **0.65** | **0.41** | **22.0** | **14.6** | **6.58** |

Table 4: Results on AV-MNIST in MSEs ($\times 10^{-4}$) across different sampling ratios.

| Modality | | Images | | | Audios | | |
|---|---|---|---|---|---|---|---|
| Range | $R^{\min}$ / $R^{\max}$ | 0.00 / 0.25 | 0.25 / 0.50 | 0.50 / 1.00 | 0.25 / 0.50 | 0.50 / 0.75 | 0.75 / 1.00 |
| Functa | | 29.7 | 7.98 | 3.84 | 2.98 | 3.03 | 3.00 |
| w/ MetaSGD | | 29.5 | 7.75 | 3.65 | 0.88 | 0.42 | 0.24 |
| w/ GAP | | 29.8 | 7.98 | 3.83 | 0.93 | 0.48 | 0.30 |
| w/ ALFA | | 27.3 | 6.84 | 3.18 | 0.78 | 0.36 | 0.20 |
| w/ MTNPs | | 29.3 | 11.9 | 7.25 | 2.36 | 2.30 | 2.22 |
| w/ Encoder | | 22.3 | 5.66 | 2.71 | 0.76 | 0.46 | 0.33 |
| w/ **MIA** | | **19.7** | **4.80** | **2.05** | **0.48** | **0.26** | **0.11** |
| Composers | | 33.4 | 9.86 | 2.77 | 1.02 | 0.47 | 0.22 |
| w/ MetaSGD | | 34.6 | 11.4 | 4.48 | 1.00 | 0.46 | 0.22 |
| w/ GAP | | 33.1 | 10.1 | 3.56 | 1.08 | 0.59 | 0.37 |
| w/ ALFA | | 30.4 | 8.19 | 2.37 | 1.04 | 0.49 | 0.22 |
| w/ MTNPs | | 29.9 | 12.0 | 7.52 | 2.57 | 2.56 | 2.52 |
| w/ Encoder | | 22.8 | 6.02 | 2.62 | 0.91 | 0.56 | 0.39 |
| w/ **MIA** | | **19.1** | **4.26** | **1.29** | **0.66** | **0.29** | **0.14** |

Figure 4: Qualitative comparisons in ERA5 and AV-MNIST. Left: Our method achieves the lowest errors (rendered in viridis color) in modeling real-world climate data. Right: Our method accurately predicts digit types with little image data, while recovering the structures of audios more faithfully.

modalities. We employ AVMNIST dataset (Vielzeuf et al., 2018), a collection of greyscale digit images (LeCun et al., 2010) and their pronounced audios (Jackson, 2016). We use the original images of a $28 \times 28$ resolution, while we trim the audios with a sampling rate of 2kHz and a duration of 1 second. We employ the original train/test split for meta-training and meta-testing, respectively. We set the sampling ratio ranges $[R_m^{\min}, R_m^{\max}]$ to $[0.001, 1.000]$ and $[0.250, 1.000]$ for images and audios.

**Results.** In Table 4, we see that Functa-based CAVIA entirely collapses in the audio modality, probably due to the high complexity in capturing high-frequency contents and noises in audios within limited optimization steps. MTNPs also fails at approximating the audio signals properly, achieving the second highest errors. We conjecture that this is due to their specialized attention and pooling mechanisms, particularly designed for multimodal signals sharing homogeneous coordinate systems and with explicit spatial alignment between modalities (Kim et al., 2022a). Unlike MTNPs, the another encoder-based method (Encoder) succeeds in modeling both signals, supporting the claims in the literature that careful parameterization can improve their fitting performances (Guo et al., 2023). Finally, our approach achieves the best performances in both modalities and all ranges of sampling ratios. Interestingly, our method predicts digit classes accurately given only one support point from an image (please see Figure 4), demonstrating its strong cross-modal generalization capability.

## 5.5 ANALYSIS STUDY

**Performances vs modules.** We investigate the roles of the three modules (*i.e.* USFTs, MSFTs, and Fusion MLPs) of SFTs in terms of both memorization and generalization performances. We construct ablated methods by gradually augmenting Composers with USFTs, MSFTs, and Fusion MLPs one at a time. We evaluate their fitting performances over the provided support sets and unobserved non-support parts separately, indicating the performances on memorization and generalization, respectively. We report the relative error reduction achieved by each method compared to the vanilla Composers, averaged across the modalities. As in the results of Table 5a, USFTs specialize in enhancing modality-specific patterns that are useful for memorization, while MSFTs further emphasize shared information for generalization. Finally, Fusion MLPs effectively integrate the representations from USFTs and MSFTs, achieving superior performances on both memorization and generalization.

Table 5: Ablation study on the components in SFTs on generalization (G) and memorization (M) ability. We report relative error reduction (↑) achieved by ablated methods over Composers.

| USFTs | MSFTs | MLPs | Syn. G | Syn. M | Celeb. G | Celeb. M | AVM. G | AVM. M |
|---|---|---|---|---|---|---|---|---|
| ✓ | ✗ | ✗ | 44.4 | 63.6 | 58.3 | 92.1 | 60.7 | 84.1 |
| ✓ | ✓ | ✗ | 86.7 | 75.5 | **61.5** | 92.5 | 63.9 | 84.6 |
| ✓ | ✓ | ✓ | **88.7** | **81.6** | 61.4 | **93.0** | **65.6** | **88.7** |

(a) Impact of USFTs, MSFTs, and Fusion MLPs.

| Params | Grads | Syn. G | Syn. M | Celeb. G | Celeb. M | AVM. G | AVM. M |
|---|---|---|---|---|---|---|---|
| ✓ | ✗ | N/A | N/A | N/A | N/A | N/A | N/A |
| ✗ | ✓ | 83.7 | 58.4 | 60.6 | 91.6 | 61.7 | 87.3 |
| ✓ | ✓ | **88.7** | **81.6** | **61.4** | **93.0** | **65.6** | **88.7** |

(b) Impact of parameters and gradients.

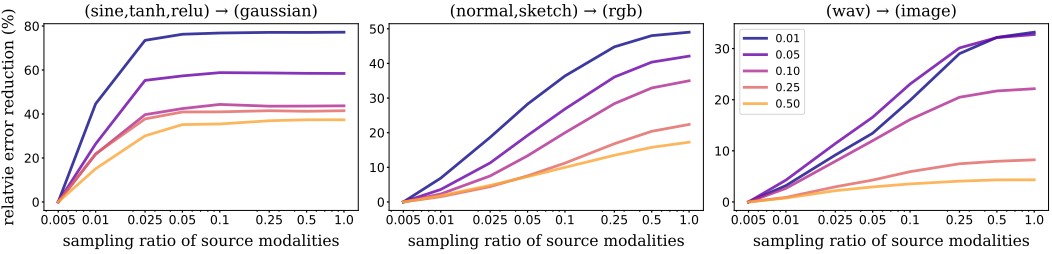

Figure 5: Relative error reduction (↑) when multimodal support set sizes increase. Colors indicate sampling ratios of the target modality, while the x-axis represents sampling ratios of the sources.

**Performances vs types of states.** We also study the impact of states utilized by SFTs, namely parameters and gradients. Similar to the previous analysis, we design two additional baselines by ablating either parameters or gradients from our full approach, and report their relative error reduction compared to Composers. Table 5b reports the results. We find the parameters-only method fails entirely (reported as N/A), while the best performance is achieved when incorporating both. This suggests that how the loss landscape is changing (*i.e.* gradients) is the most crucial information while incorporating the current knowledge of the learners (*i.e.* parameters) as well further boosts the capture of the underlying structures of multimodal signals and their associated optimization landscapes.

**Performances vs sizes of the multimodal contexts.** We validate whether our SFTs can (1) correctly identify valuable cross-modal patterns in learners' states and (2) utilize them to benefit the learners. For the first aspect, we examine how attention patterns of MSFTs relate to the quality of gradients. We calculate the Pearson correlation coefficient, represented as $C_{ij}$, which reflects the relationship between attention weights assigned to the learner's state representations for modality $i$ and the support set size for modality $j$. We use support set size as a proxy for gradient quality since the amount of gradient noises often relate to the size of the support sets (Zhang et al., 2019). Notably, as presented in Table 9, we observe strong positive correlations along the diagonal, suggesting that MSFTs refrain from updating the state representations for a specific modality when observations within that modality suffice. Conversely, the strong negative correlations off the diagonal imply that MSFTs compensate for suboptimal gradient quality in one modality by directing more attention to other modalities.

To examine the second aspect, we assess the performances on one target modality while the sizes of multimodal support sets from other modalities are varied. As in previous experiments, we report the relative error reduction by our method compared to the one obtained without the supports from the other modalities. Figure 5 visualizes the results that, in general, increased availability of multimodal support sets contributes to better performances in the target modality, while the benefit gained from multimodal observations becomes more significant when the size of the target support sets is smaller. Based on these investigations, we confirm the model's ability to correctly identify and leverage cross-modal interactions, signifying its potential for applications where data constraints often prevail.

## 6 CONCLUSION

We introduced a new gradient-based meta-learning framework for INRs, resulting in substantial improvements in the convergence and performance of existing meta-learning INR frameworks. It was primarily attributed to our novel Multimodal Iterative Adaptation (MIA), which empowers separate INR learners to collaborate and exchange their learning progress or knowledge through State Fusion Transformers (SFTs). This unique adaptation scheme harnesses the cross-modal interactions among the states of the learners, facilitating rapid convergence and enhancing model performance.

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

APPENDIX

This section provides more results and details that could not be included in the main paper.

## A    MORE QUALITAIVE RESULTS

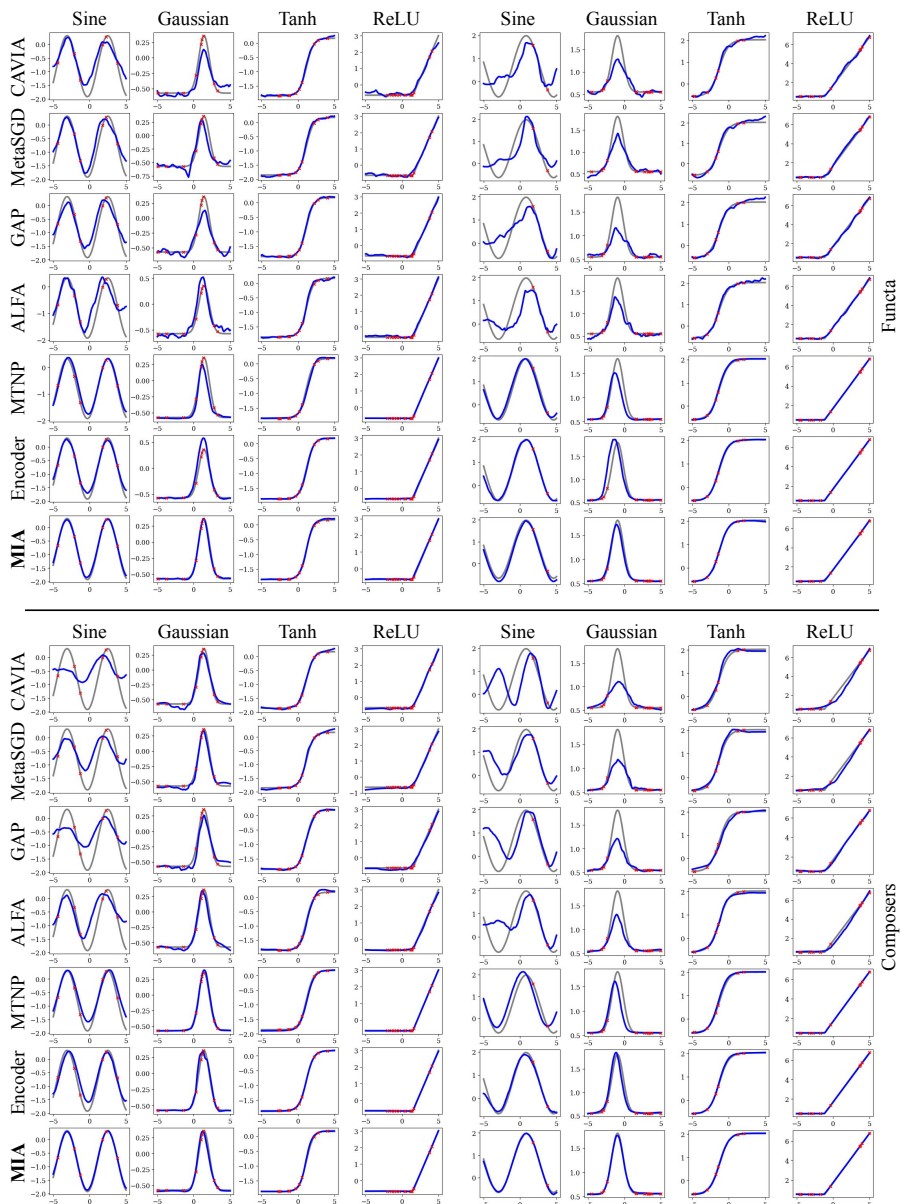

Figure 6: Qualitative comparisons on 1D synthetic functions. The black and blue lines represent the ground-truth and approximated signals recovered by each method, respectively, while red crossmarks pinpoint the locations of the provided support points. While all the methods operate well in relatively smooth and low-curvature signals (*e.g.* Tanh and ReLU), the baselines either fail entirely in Sine and Gaussian modalities (*i.e.* the unimodal approaches) or struggle to approximate them correctly (*i.e.* the multimodal baselines). Unlike the baselines, our MIA fits almost perfectly to all functions, verifying its capability in fusing multiple source of information to improve the performances.

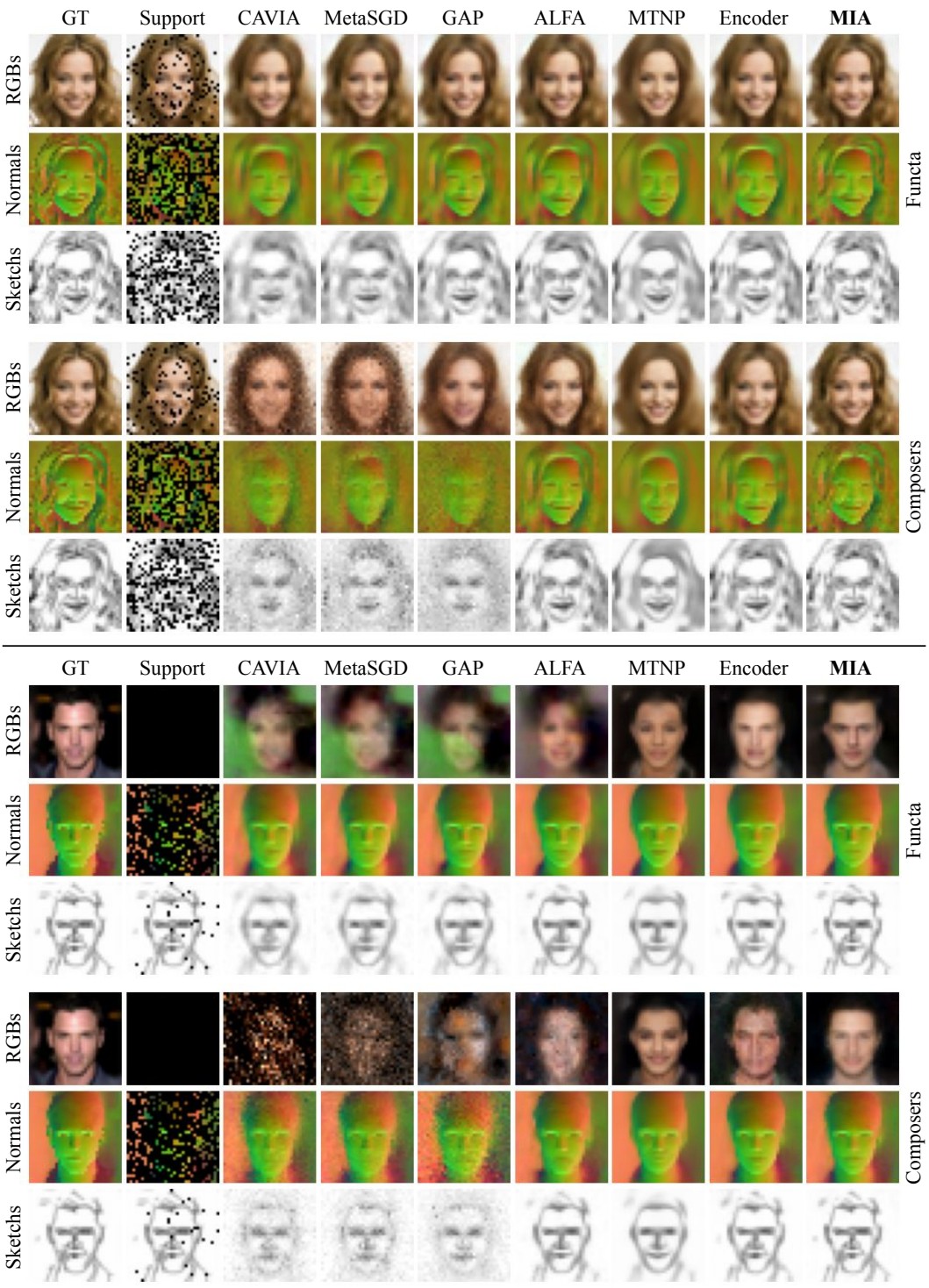

Figure 7: Qualitative comparisons on CelebA 2D visual modalities. Compared to the baselines, our MIA can render high-frequency details and nuances contained in each modality signal thanks to its enhanced convergence and memorization capability (upper pane). Even in the absence of sufficient data, our method can generalize well via strong ability in cross-modal generalization (lower pane).

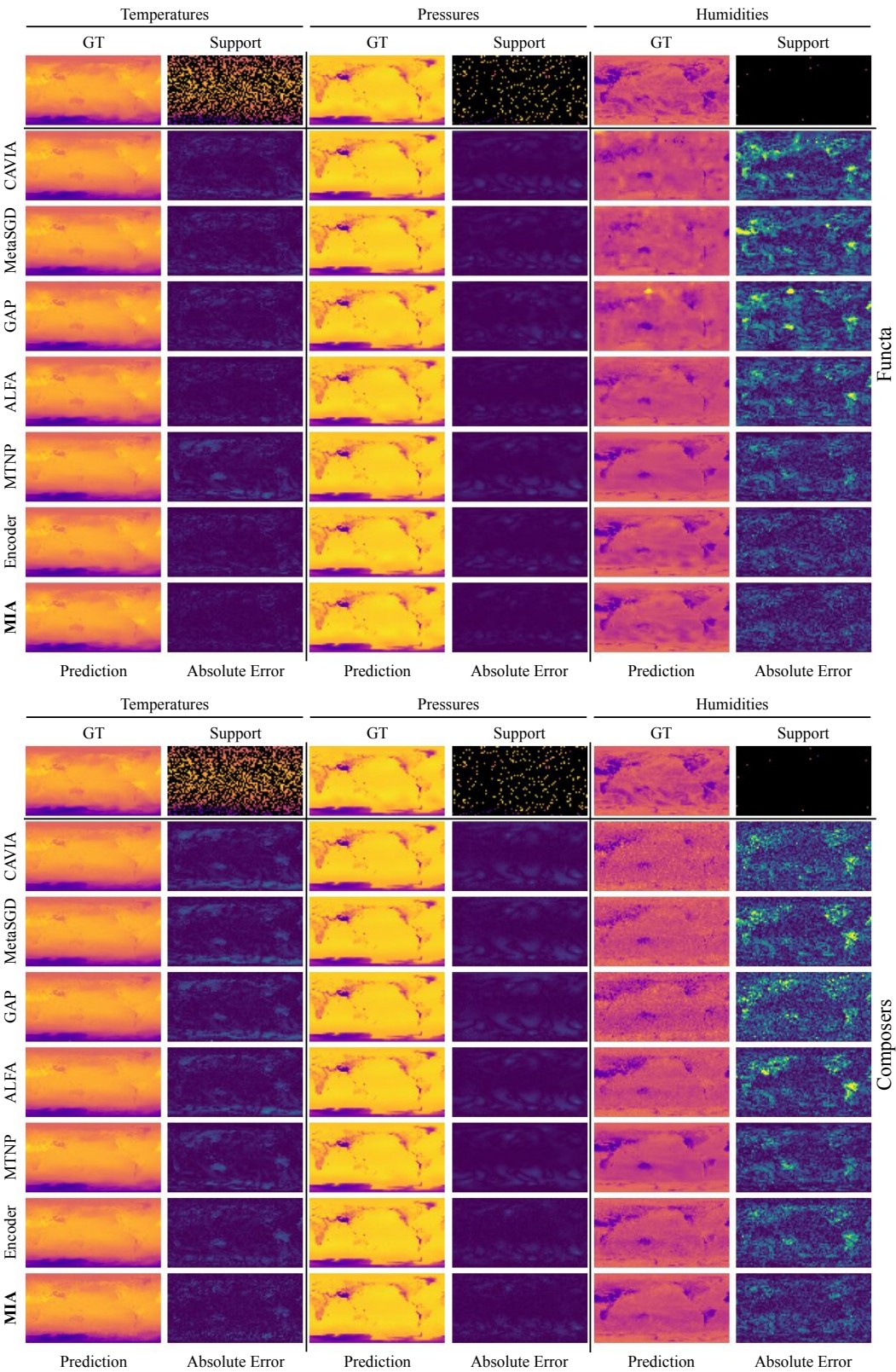

Figure 8: Qualitative comparisons on ERA5 global climate data. The first row of each pane indicates the grount-truth signals and provided support points, while the rest represents the approximations and their absolute errors achieved by each method. Our MIA shows superior generalization capability than the existing baselines, demonstrating its versatility in applications for real-world climate estimation.

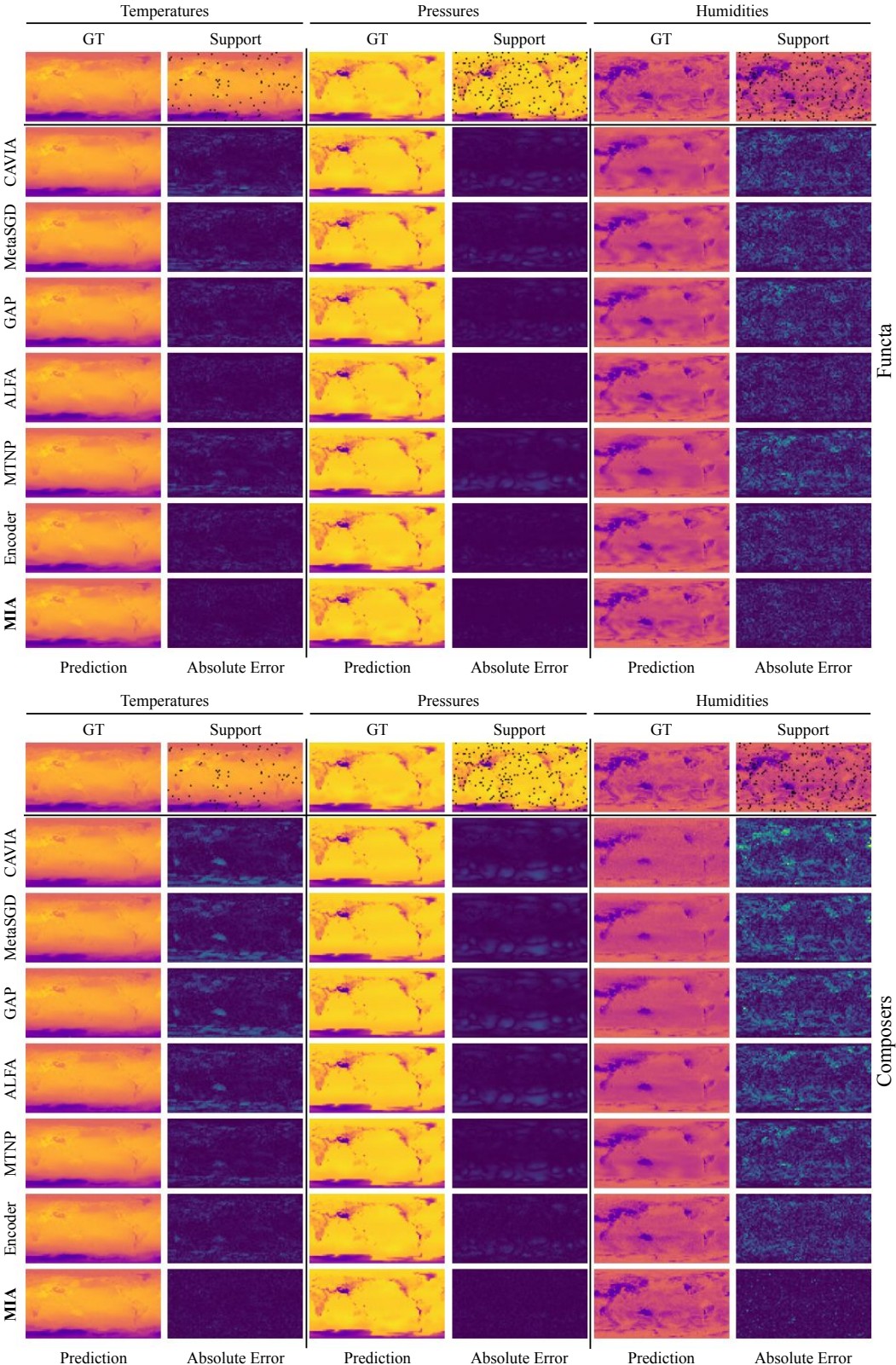

Figure 9: Qualitative comparisons on ERA5 global climate data. The first row of each pane indicates the grount-truth signals and provided support points, while the rest represents the approximations and their absolute errors achieved by each method. Our MIA shows superior convergence speed than the existing baselines, which struggle to approximate the climate data precisely and show high errors even when abundant observations are available.

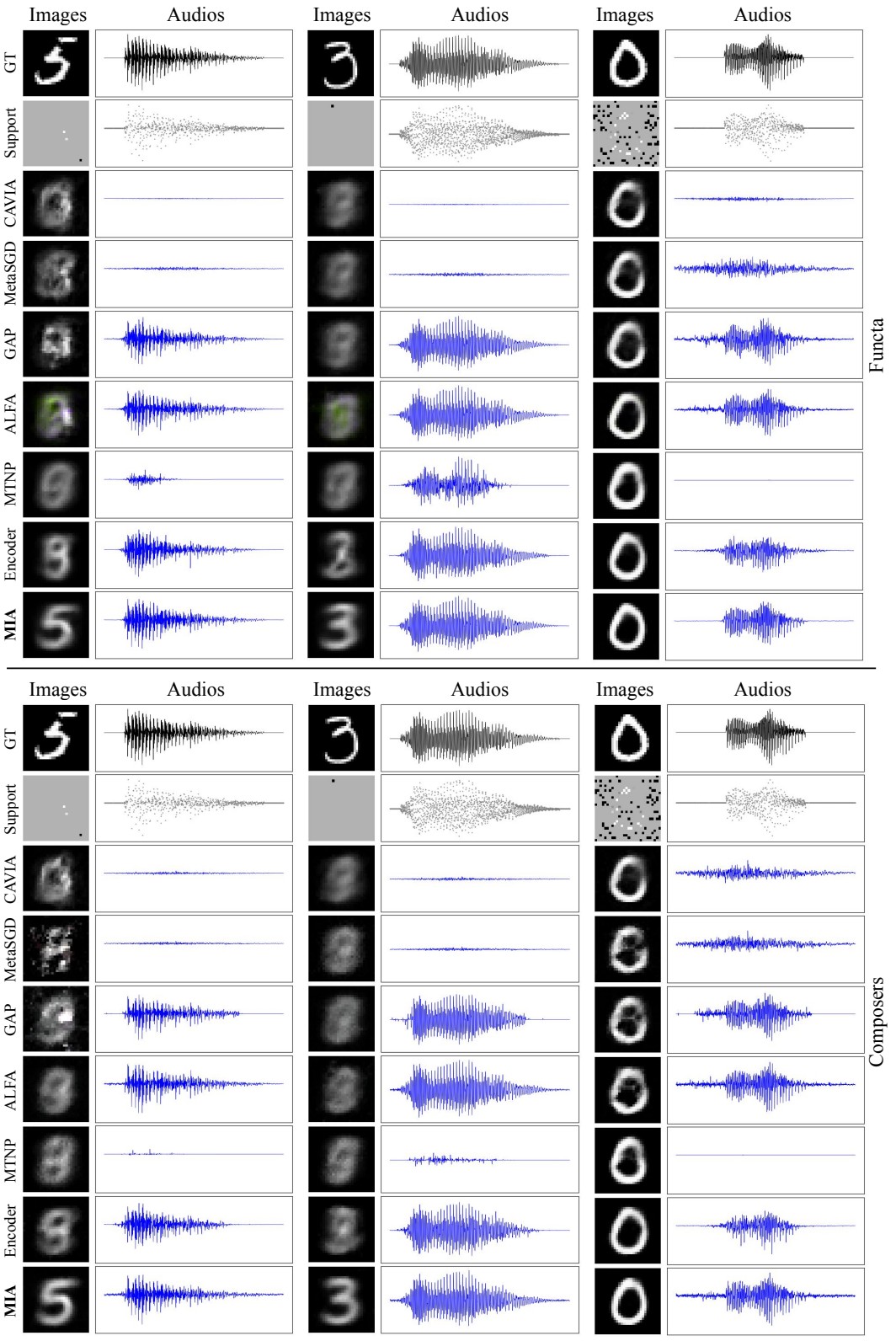

Figure 10: Qualitative comparisons on AV-MNIST. Our method successfully infer the digit classes from audio signals when little data is available from images (left two columns), while enjoying superior fitting performances when sufficient data is available (rightmost column).

# B    Details of the INR frameworks

In this work, we adopt the meta-learning frameworks for INRs proposed in Functa (Dupont et al., 2022a; Bauer et al., 2023) and Composer (Kim et al., 2023). They builds upon CAVIA (Zintgraf et al., 2018), where two separate parameters are meta-learned: the INR parameters $\theta$ and the context parameters $\phi \in \mathbb{R}^{S \times D_\phi}$. In particular, the parameters $\theta$ of INRs are meta-learned to capture underlying structure of data or data-agnostic shared information across signals. In addition, the context parameters $\phi \in \mathbb{R}^{S \times D_\phi}$, a set of $D_\phi$-dimensional features, are adapted to each signal and encode per-signal variations or characteristics, which are then utilized to condition the parameters of INRs to recover the signals they are modeling. Besides the shared concepts, they differ in how they construct and utilize the context parameters to condition the INR parameters, which we detail below.

**Functa.** In Functa, the context parameters $\phi_n \in \mathbb{R}^{S \times D_\phi}$ for each signal are constructed as a spatially-arranged grid of $D_\phi$-dimensional features, where each of them encodes local variations in a signal. This grid of features is then utilized to modulate the activations of each INR layer. To do so, the set of features is first processed by an additional linear convolutional module as $\psi_n = f_{\text{conv}}(\phi_n; \theta_{\text{conv}})$, followed by bilinear/trilinear interpolation to compute a layer-wise affine transformation parameters $\psi_n(x) = \texttt{Interp}(x; \psi_n)$ that scale and/or shift the activations of each layer in INRs given an input coordinate $x$. In the original paper, the authors of Functa opt to adopt a shift-only modulation scheme since it shows better rate-distortion (compression vs performance) trade-offs than using both. In contrast, we adopt a scale-only modulation scheme since it is empirically shown to perform slightly better in our earlier experiments. Also, we do not modulate the activations of the first and last layers since we empirically find it stabilizes the training without hurting the performances.

**Composers.** Unlike Functa that adopt local grid of features and FiLM-like modulation scheme (Perez et al., 2018) applied to multiple layers of INRs, Composers introduce a set of non-local features $\mathbf{V}_n \in \mathbb{R}^{S \times D_\phi}$ that are used as a low-rank approximation $\mathbf{W}_n = \mathbf{U}\mathbf{V}_n$ on parameters $\mathbf{W}_n$ of a single layer in INRs. Here, $\mathbf{U} \in \mathbb{R}^{S \times D_\phi}$ is another low-rank approximation matrix that is incorporated in the INR parameters $\theta$ and captures global patterns shared across various signals, which are composed and modulated in a data-specific manner via $\mathbf{V}_n$. Following the original work (Kim et al., 2023), we approximate the second layer of INRs using $\mathbf{U}$ and $\mathbf{V}_n$.

Throughout all experiments, we use a 5-layer MLPs with 128 hidden dimensions and ReLU non-linearities in-between as a base INR architecture for both Functa and Composers. We also use random fourier features (Tancik et al., 2020) with $\sigma = 30$ to encode positional information, except that we do not use these features for the experiments on the 1D synthetic dataset.

# C    Details on State Fusion Transformers (SFTs)

We construct USFTs and MSFTs using the transformer block of ViT (Dosovitskiy et al., 2021), where each of them is parameterized by one transformer block (*i.e.* $L_1 = L_2 = 1$) with a width of 192, a MLP dimension of 192, and 3 attention heads. In addition, we set the dimension of state representations $z_{nm}^{(k)} \in \mathbb{R}^{S_m \times D_z}$ to $D_z = 192$ for both USFTs and MSFTs in all experiments.

To compute the state representations $z_{nm}^{(k)}$ for each modality, we first apply LayerNorm (Ba et al., 2016) to the parameters $\phi_{mn} \in \mathbb{R}^{S_m \times D_\phi}$ and the gradients $g_{mn} \in \mathbb{R}^{S_m \times D_\phi}$ separately to stabilize the training. Then, we concatenate them along the feature dimension (*i.e.* $[\phi_{mn} \,\|\, g_{mn}] \in \mathbb{R}^{S_m \times 2D_\phi}$), followed by projecting them to the hidden space of USFTs and obtain the state representations $z_{nm}^{(k)}$ via a two-layer MLPs with a hidden dimension of 192 and ReLU activations, which is dedicated to each modality. In addition, we add positional embedding to the state representations before feeding them into USFTs, where we use sinusoidal positional encodings (Mildenhall et al., 2020) and learnable embedding for Functa and Composers, respectively. Finally, we parameterize Fusion MLPs with a 3-layer MLPs with a hidden dimension of 192 and ReLU non-linear activations. Also, we embed LayerNorm into the first and the penultimate layers of Fusion MLPs to stabilize the training.

We also attach PyTorch-style pseudo code for MIA in Listing 1 and SFTs in Listing 2, respectively.

```python
def inner_optimization_step(inr_model_dict, sft_model, modes,
                            ctx_params_dict, x_spt_dict, y_spt_dict):
    # inr_model_dict: A dictionary of Pytorch modules of INRs for each modality.
    # sft_model: Pytorch module for State Fusion Transformers.
    # modes: A list of modalities.
    # ctx_params_dict: a dictionary of the context parameters of INRs for each modality.
    # x_spt_dict: A dictionary of the provided support coordinates for each modality.
    # y_spt_dict: A dictionary of the provided support features for each modality.
    grad_dict = dict()
    for mode in modes:
        # for each modality, predict features `y` with given support cooridnates `x`.
        y_pred = inr_model_dict[mode](x_spt_dict[mode], ctx_params_dict[mode])
        loss = mse_loss(y_pred, y_spt_dict[mode])
        # compute gradients w.r.t context parameters.
        grad_dict[mode] = torch.autograd.grad(
            loss,
            ctx_params_dict[mode],
            create_graph = True # Set to `True` during meta-training, otherwise `False`.
        )[0]
    # compute enhanced weight updates via SFTs.
    grad_dict = fuse_states(sft_model, modes, grad_dict, ctx_params_dict)
    for mode in modes:
        # update context parameters using the enhanced gradients for each modality.
        ctx_params_dict[mode] = ctx_params_dict[mode] - grad_dict[mode]

    return ctx_params_dict
```

Listing 1: PyTorch style pseudo-code for inner optimization step via MIA.

```python
def fuse_states(sft_model, modes, grad_dict, ctx_params_dict):
    # sft_model: Pytorch module for State Fusion Transformers.
    # modes: A list of modalities.
    # grad_dict: a dictionary of the gradients w.r.t the context parameters for each modality.
    # ctx_params_dict: a dictionary of the context parameters of INRs for each modality.
    ori_state_dict = dict()
    uni_state_dict = dict()
    multi_state_dict = dict()
    states = []
    for mode in modes:
        # project gradients and context params to the hidden space
        ori_state_dict[mode] = sft_model.proj_mlp[mode](grad_dict[mode],
                                                         ctx_params_dict[mode])
        # add positional embedding and apply USFT for each modality
        state = ori_state_dict[mode] + sft_model.pos_emb[mode]
        uni_state_dict[mode] = sft_model.usft[mode](state)
        states.append(uni_state_dict[mode])
    # concat states for all modalities and apply MSFT
    states = torch.cat(states, dim = 1)
    states = sft_model.msft(states)
    for mode in modes:
        # calculate enhanced gradients by applying FusionMLP for each modality
        multi_state_dict[mode] = split(states, mode)
        state = torch.cat([ori_state_dict[mode],
                           uni_state_dict[mode],
                           multi_state_dict[mode]], dim = -1)
        grad_dict[mode] = sft_model.fusion_mlp[mode](state)

    return grad_dict
```

Listing 2: PyTorch style pseudo-code for State Fusion Transformers (SFTs).

Table 6: List of common configurations for each dataset.

| Hyperparameters | | Synthetic | CelebA | ERA5 | AVMNIST |
|---|---|---|---|---|---|
| modalities | | Sine, Gaussian Tanh, Relu | RGB, Normal Sketch | Temperature Pressure Humidity | Images Audios |
| number of data | train | 900 | 27143 | 11327 | 60000 |
| | test | 100 | 2821 | 2328 | 10000 |
| batch size | | 64 | 32 | 16 | 32 |
| epoch | | 16,000 | 300 | 300 | 300 |
| resolution | | (200, 1) | RGB - (32, 32, 3) Normal - (32, 32, 3) Sketch - (32, 32, 1) | (46, 90, 1) | Images - (28, 28, 1) Audios - (2000, 1) |
| sampling range $[R^{\min}, R^{\max}]$ | | [0.01, 0.1] | [0.001, 1] | [0.001, 1] | Images - [0.001, 1] Audios - [0.250, 1] |
| outer learning rate | | $10^{-4}$ | | | |
| momentum $(\beta_1, \beta_2)$ for Adam | | (0.9, 0.999) | | | |
| total inner step $K$ for optimization-based methods | | 3 | | | |
| scale for uncertainty lr | | 1 | 1 | 0.1 | 10 |
| width/depth of INRs | | 128/5 | | | |
| dimension of context parameters $\phi$ | Functa | (8, 16) | (8, 8, 16) | (8, 16, 16) | Images - (8, 8, 4) Audios - (64, 32) |
| | Composer | (8, 128) | (64, 128) | (128, 128) | Images - (64, 128) Audios - (64, 128) |
| $\sigma$ for fourier feature | | None | 30.0 | 30.0 | 30.0 |
| bootstrapping factor for evaluation | | 10 | 4 | 4 | 4 |

## D    MORE EXPERMINENTAL DETAILS AND RESULTS

### D.1    COMMON DETAILS

We subsample coordinate-feature pairs from the full set during the meta-training phase to promote the generalization, as well as memorization. For each signal $\mathcal{D}_{nm}$, the sampling ratio $R_{nm}$ is independently and identically drawn from a uniform distribution $\mathcal{U}(R_m^{\min}, R_m^{\max})$ for each modality $m$ to construct the support sets $\mathcal{D}_n^{\text{train}}$. For evaluation, we employ a bootstrapping technique on the meta-test dataset. We iteratively sample the support and query sets multiple times for each data to mitigate potential variances that may arise from the sampling process. For 1D synthetic experiments, we apply a bootstrapping factor of 10, while for other scenarios, the factor is set to 4. In all experiments, we use Adam optimizer (Kingma & Ba, 2014) for meta-optimization, with a learning rate of $10^{-4}$ and the momentum parameters are set as $(\beta_1, \beta_2) = (0.9, 0.999)$.

Also, we apply uncertainty-aware loss weighting technique (Kendall et al., 2018) for all multimodal methods. Please find Table 6 for the clear list of common configurations for each dataset.

### D.2    BASELINES

**CAVIA.** We use a global fixed learning rate of $\alpha = 1.0$ to adapt the context parameters of CAVIA-like methods (Functa and Composers) in the inner-loop for all experiments.

**MetaSGD.** Li et al. (2017) propose to use a meta-learned per-parameter learning rate $\alpha \in \mathbb{R}^{S \times D_\phi}$ in the inner-loop adaptation phase, which is optimized along with the meta-learner in the outer-loop

optimization phase. We apply this technique to adapt the context parameters of CAVIA-based frameworks, where we initialize their initial values to 1.0 in all experiments.

**GAP.** Kang et al. (2023) propose to accelerate the optimization process via Geometry-Adaptive Preconditioner (GAP), which preconditions the gradients $\mathbf{G}^{(k)}$ at inner step $k$ by manipulating its singular values with meta-learned parameters $\mathbf{M}$. The procedure can be written as:

$$\tilde{\mathbf{G}}^{(k)} = \mathbf{U}^{(k)}(\mathbf{M} \cdot \mathbf{\Sigma}^{(k)})\mathbf{V}^{(k)^{\mathrm{T}}}, \text{ where } \mathbf{G}^{(k)} = \mathbf{U}^{(k)}\mathbf{\Sigma}^{(k)}\mathbf{V}^{(k)^{\mathrm{T}}}. \tag{13}$$

In the original paper (Kang et al., 2023), gradient matrix unfolding technique is introduced to facilitate SVD on the gradients of convolutional weights. Different from the original setup, now the shape of the context parameters and their associated gradient matrices is $\mathbb{R}^{S \times D_\phi}$. Therefore, we do not using this unfolding technique and define the meta parameters as $\mathbf{M} = \mathrm{diag}(\mathrm{Sp}(M_1), \ldots \mathrm{Sp}(M_{\min(S,D_\phi)}))$.

In addition, we experiment with *Approximate GAP* as well and report the best performances achieved from the two methods. This approximated version bypasses the need of SVD for calculating the preconditioned gradients by $\tilde{\mathbf{G}}^{(k)} \simeq \mathbf{M}\mathbf{G}^{(k)}$.

**ALFA.** ALFA is proposed to meta-learn the weight update procedure along with the weights to facilitate the learning. Unlike GAP, ALFA introduces an additional meta-learned neural network $h(\phi^{(k)}, g^{(k)}; \xi)$ that dynamically predicts learning rates $\alpha^{(k)}$ and weight decaying terms $\beta^{(k)}$ in a data-specific manner and for each inner step $k$, given the current parameters $\phi^{(k)}$ and their gradients $g^{(k)}$ of the learners. The resulting weight update rule can be described as follows:

$$\phi^{(k+1)} = \beta^{(k)} \cdot \phi^{(k)} - \alpha^{(k)} \cdot g^{(k)}, \quad \text{where } g^{(k)} = \nabla_{\phi^{(k)}}\mathcal{L}^{(k)}. \tag{14}$$

To construct the meta-learned network $h$, we follow the original setup (Baik et al., 2023) and parametrize it with a 2-layer MLPs with ReLU activations. Also, we reduce the context parameters and gradients by averaging them along the feature dimensions (*i.e.* $g, \phi \in \mathbb{R}^{S \times D_\phi} \to \bar{g}, \bar{\phi} \in \mathbb{R}^S$) and feed them into the meta-learned network $h$. Finally, we augment the predicted learning rates and decaying terms $\alpha^{(k)}, \beta^{(k)} \in \mathbb{R}^S$ with additional meta-learned weights $\alpha_0, \beta_0 \in \mathbb{R}^S$, as suggested in the paper. We set the initial values of $\alpha_0, \beta_0$ to 1.

**MTNP.** Multitask Neural Processes (MTNPs) (Kim et al., 2022a) is another class of meta-learning appproach for INRs based on Neural Processes (Garnelo et al., 2018), aimed at modeling multimodal signal distributions, similar to ours. This method replaces iterative optimization steps with feed-forward encoder networks to directly predict the parameters of INRs from observed signals. For this, MTNPs adopts a dual-stream and hierarchical fusion approach to capture cross-modal relationships among signals. The first stream, driven by a latent encoder, is tasked to capture uncertainty in recovering the entire function given partial observations and to infer global latent variables shared by the observed data points for each signal in a modality ($\mathcal{D}_{mn}$). On the other hand, the second stream is guided by a deterministic encoder and is responsible for extracting local per-coordinate representations specific to each data point $(x_{nm}, y_{nm})$ that belongs to a signal in a modality. In addition, to improve the expressive power of the model, each stream is composed of a stack of specialized hierarchical multi-head attention blocks, where the earlier part captures the dependencies among data points within a modality and the latter discover their potential cross-modal relationships. For instance, in the first latent stream, cross-modal relationships among global latent variables for each modality are captured. Similarly, multimodal dependencies among local representations that belong to the same coordinate are considered in the second deterministic stream. We refer more interested readers to Section 3 and Figure 2 in the original paper of MTNPs (Kim et al., 2022a).

We apply two modifications to MTNPs to adapt it in our experiments: (1) We change the output dimension of the latent encoder so that it directly predicts the context parameters $\phi_{nm} \in \mathbb{R}^{S_{nm} \times D_\phi}$ that conditions the parameters of INRs. (2) For experiments on AVMNIST, we omit the module in the deterministic stream that captures cross-modal interactions among the axis-aligned local representations since there is no one-to-one correspondence between the image and audio coordinates.

**Encoder.** We construct another encoder-based method that adopts the architecture design of SFTs for the multimodal encoders. This multimodal encoder is composed of four parts: (1) a cross-attention block that aggregates the observed data points ($\mathcal{D}_{nm}^{\mathrm{train}}$) for each modality and extract signal representations $z_{nm} \in \mathbb{R}^{S_m \times D_z}$ from them, (2) a unimodal transformer block that captures and enhances potential dependencies within each modality representation, (3) a multimodal transformer block that further captures cross-modal interactions among the representations from different modalities and

improves the representations, and (4) a projection MLPs that predicts the context parameters $\phi_{nm}$ of INRs for each modality. Here the unimodal and multimodal transformer blocks share the same architecture design with our USFTs and MSFTS, respectively.

# E    COMPARISONS WITH EXISTING MULTIMODAL META-LEARNING STUDIES

Multimodal meta-learning is not new in the domain of meta-learning. However, existing studies (Vuorio et al., 2018; 2019; Abdollahzadeh et al., 2021; Sun et al., 2023) differ significantly from our work in terms of the notion of modality and problem setups. This not only makes direct comparisons challenging but also hinders fair comparisons since evaluating their frameworks or ours to within each other's context requires substantial modifications in the methodologies. Nonetheless, in this section, we describe the problem setup and the method focused by these studies, followed by their comparative evaluation results in the context of our joint multimodal signal modeling scenarios.

**MMAML & KML**. Unlike the prevalent focus of meta-learning and meta-testing on single-domain problems, exemplified by few-shot classification tasks on a single individual dataset like Omniglot, mini-Imagenet, or CUB datasets, works such as Vuorio et al. (2018; 2019) and Abdollahzadeh et al. (2021) extend their scope to encompass multiple domains of datasets. For example, they explore simple regression problems across a union of sinusoidal, linear, and tanh functions, few-shot classification tasks combining Omniglot, mini-Imagenet, and CUB datasets, or reinforcement learning scenarios across various environments such as Point Mass, Reacher, and Ant. In these studies, each domain, whether dataset or environment, is treated as a distinct modality, leading to their concept of multimodal meta-learning. Importantly, this concept of multimodality differs from the traditional understanding related to data types, as explicitly clarified in Section 3 of Abdollahzadeh et al. (2021).

These works highlight that meta-learning a single initialization may be suboptimal due to the multimodal nature of the problem. To address this, they introduce an additional meta-learned module known as a task encoder. The role of this task encoder is to identify the latent modality of the observed data and predict modulation parameters. These parameters guide the learned single initialization towards modality-specific initializations. Experimental results indicate that the task encoder indeed learns to identify the modality (or dataset domain), and consequently, adapting from modality-specific initializations yields better performance than relying on a modality-agnostic single initialization.

However, it's important to note that in these papers, each data point is assumed to be sampled iid from the union of datasets. As a result, the learner adapts independently to each data point, without explicitly leveraging cross-modal relationships among modalities or incorporating mechanisms for multimodal fusion. For these reasons, we categorize these approaches as inherently unimodal meta-learning methods.

Upon close inspection for enabling a fair comparative evaluation on these approaches, we apply the following modifications to our Encoder baseline to construct MMAML (Vuorio et al., 2018; 2019): (1) we remove the multimodal transformer in Encoder, (2) share its unimodal encoders and INR parameters across all modalities so that the framework operates on a per-modality basis, (3) and allow the estimated context parameters by the unimodal encoders to be optimized for $K$ steps, similar to the other optimization-based approaches.

**AMML**. Unlike the previously mentioned studies, Sun et al. (2023) delve into a scenario where multimodality relates specifically to data types. Their primary focus lies in addressing multimodal sentiment analysis problems, where the learner is tasked with classifying discrete sentiment scores ranging from 1 to 7 or binary sentiment classes (positive or negative). Since this classification relies on diverse sources of information represented in different modalities, encompassing texts, images, and audios, the task is referred to as multimodal inference in their work.

In their work, the authors underscore the limitations of existing methodologies, pointing out their inability to consider the heterogeneous convergence properties of each unimodal encoder network. Naively adapting these encoders using identical optimization algorithms (*e.g.* SGD with the same learning rate) results in suboptimal unimodal encoder networks. Consequently, the fusion of these suboptimal representations yields unsatisfactory outcomes for the final prediction.

To address this, Sun et al. (2023) propose a meta-learning approach involving the independent adaptation of each modality-specific encoder using the unimodal classification loss within the inner

Table 7: Quantitative comparisons on the multimodal 1D synthetic functions. We report the normalized MSEs ($\times 10^{-2}$) computed over distinct ranges of sampling ratios, averaged over 5 random seeds.

| Modality | | Sine | | | Gaussian | | | Tanh | | | ReLU | | |
|---|---|---|---|---|---|---|---|---|---|---|---|---|---|
| Range | $R^{\min}$ | 0.01 | 0.02 | 0.05 | 0.01 | 0.02 | 0.05 | 0.01 | 0.02 | 0.05 | 0.01 | 0.02 | 0.05 |
| | $R^{\max}$ | 0.02 | 0.05 | 0.10 | 0.02 | 0.05 | 0.10 | 0.02 | 0.05 | 0.10 | 0.02 | 0.05 | 0.10 |
| Functa | | 44.26 | 16.07 | 3.319 | 18.81 | 4.388 | 0.953 | 22.61 | 3.667 | 0.586 | 65.29 | 10.79 | 2.157 |
| w/ MMAML | | 43.35 | 10.09 | 1.233 | 30.87 | 4.270 | 0.698 | 17.20 | 2.351 | 0.312 | 72.79 | 5.630 | 0.261 |
| w/ AMML | | 10.12 | 4.462 | 1.717 | 1.253 | 0.959 | 0.638 | 1.719 | 0.541 | 0.267 | 7.063 | 1.254 | 0.305 |
| w/ **MIA** | | **6.386** | **2.058** | **0.547** | **1.057** | **0.571** | **0.281** | **1.285** | **0.378** | **0.131** | **5.069** | **1.012** | **0.115** |
| Composers | | 37.40 | 16.70 | 5.284 | 5.923 | 3.149 | 1.460 | 14.81 | 4.053 | 1.029 | 48.49 | 11.98 | 3.232 |
| w/ MMAML | | 44.26 | 11.63 | 1.570 | 31.09 | 4.664 | 0.895 | 16.46 | 2.330 | 0.309 | 74.75 | 5.845 | 0.294 |
| w/ AMML | | 14.57 | 9.549 | 7.706 | 1.699 | 1.462 | 1.086 | 2.411 | 1.021 | 0.725 | 10.19 | 2.340 | 0.907 |
| w/ **MIA** | | **5.564** | **1.844** | **0.627** | **0.975** | **0.528** | **0.237** | **1.257** | **0.343** | **0.128** | **4.715** | **0.943** | **0.156** |

Table 8: Quantative comparisons on the multimodal 2D CelebA image function regression. We report MSEs ($\times 10^{-3}$) computed over distinct ranges of sampling ratios, averaged over 5 random seeds.

| Modality | | RGBs | | | | Normals | | | | Sketches | | | |
|---|---|---|---|---|---|---|---|---|---|---|---|---|---|---|
| Range | $R^{\min}$ | 0.00 | 0.25 | 0.50 | 0.75 | 0.00 | 0.25 | 0.50 | 0.75 | 0.00 | 0.25 | 0.50 | 0.75 |
| | $R^{\max}$ | 0.25 | 0.50 | 0.75 | 1.00 | 0.25 | 0.50 | 0.75 | 1.00 | 0.25 | 0.50 | 0.75 | 1.00 |
| Functa | | 13.44 | 4.117 | 3.052 | 2.577 | 5.067 | 2.448 | 2.092 | 1.928 | 13.29 | 7.065 | 5.704 | 5.209 |
| w/ MMAML | | 11.22 | 3.775 | 2.772 | 2.264 | 4.311 | 2.266 | 1.872 | 1.672 | 10.96 | 6.021 | 4.452 | 3.751 |
| w/ AMML | | 8.110 | 2.909 | 1.853 | 1.334 | 3.341 | 1.689 | 1.243 | 1.000 | 8.192 | 4.332 | 2.278 | 1.124 |
| w/ **MIA** | | **6.946** | **2.563** | **1.627** | **1.135** | **2.979** | **1.530** | **1.118** | **0.869** | **7.667** | **4.042** | **2.142** | **1.011** |
| Composers | | 26.40 | 15.83 | 14.30 | 13.21 | 6.979 | 4.830 | 4.630 | 4.517 | 17.99 | 14.36 | 13.27 | 12.86 |
| w/ MMAML | | 12.35 | 4.228 | 2.806 | 2.029 | 4.902 | 2.567 | 2.028 | 1.725 | 11.29 | 5.757 | 3.646 | 2.521 |
| w/ AMML | | 11.28 | 5.195 | 3.971 | 3.241 | 4.209 | 2.336 | 1.745 | 1.339 | 9.813 | 5.084 | 2.727 | 1.319 |
| w/ **MIA** | | **9.764** | **3.418** | **1.913** | **1.017** | **3.749** | **1.763** | **1.062** | **0.526** | **9.505** | **4.708** | **2.336** | **0.855** |

loop. Following this inner-loop adaptation phase, a multimodal fusion network integrates the extracted representations derived from these individually adapted unimodal encoders. Finally, the fused representations are utilized for predicting class labels for sentimental analysis. It's also worth mentioning that Transformer structures are employed for independent modality-specific encoder (such as the pre-trained BeRT for the text encoder and Vanilla Transformers for image and acoustic encoders), whereas they are not utilized for multimodal fusion.

Since their framework is not directly applicable to our joint multimodal function regression scenarios, we investigat the effectiveness of their adaptation-in-the-inner-loop followed by fusion-in-the-outer-loop scheme by applying our method's MIA only in the final optimization step.

**Results.** Each compared method is run 5 times with different random seeds on Synthetic and CelebA datasets and their results are averaged. The results are in Table 7 and 8. From the tables, we conclude the following things: (1) MMAML greatly improves the memorization performances of CAVIA (Functa and Composer) thanks to the task encoder network, while it fails to generalize better than CAVIA due to its inherently unimodal nature of MMAML. (2) AMML further improves the generalization performances of CAVIA thanks to its multimodal-fusion-in-the-outer-loop scheme. However, its performances still fall short than our MIA, demonstrating the effectiveness of joint multimodal iterative adaptation of the learners during the adaptation stages.

Table 9: Pearson Correlation Coefficient ($C_{ij}$) between the attention scores of MSFTs assigned to the learner's state representations for modality $i$ and the support set size for modality $j$.

| $C$ | Functa | | | | Composers | | | |
|---|---|---|---|---|---|---|---|---|
| | Sine | Gaussian | Tanh | ReLU | Sine | Gaussian | Tanh | ReLU |
| Sine | 0.561 | -0.393 | -0.434 | -0.357 | 0.620 | -0.520 | -0.455 | -0.523 |
| Gaussian | -0.089 | 0.290 | -0.074 | -0.020 | -0.406 | 0.473 | -0.262 | -0.222 |
| Tanh | -0.163 | -0.068 | 0.319 | -0.036 | -0.493 | -0.439 | 0.597 | -0.421 |
| ReLU | -0.144 | -0.082 | -0.165 | 0.270 | -0.265 | -0.226 | -0.117 | 0.353 |

(a) Coefficients on multimodal 1D synthetic functions.

| $C$ | Functa | | | Composers | | |
|---|---|---|---|---|---|---|
| | RGBs | Normals | Sketchs | RGBs | Normals | Sketchs |
| RGBs | 0.779 | -0.645 | -0.579 | 0.518 | -0.807 | 0.434 |
| Normals | -0.794 | 0.843 | -0.381 | -0.830 | 0.849 | -0.142 |
| Sketchs | -0.923 | -0.891 | 0.973 | -0.859 | -0.254 | 0.882 |

(b) Coefficients on multimodal 2D CelebA images.

# F    MORE ANALYSIS ON MIA

## F.1    CORRELATION BETWEEN ATTENTION PATTERN AND OBSERVATION QUALITY

Table 9 presents Pearson Correlation Coefficient ($C_{ij}$) between the attention scores of MSFTs assigned to the learner's state representations for modality $i$ and the support set size for modality $j$. The table reveals the internal mechanism on when and how SFTs facilitate the interaction among the learners, as discussed in the main paper. In this section we provide the methodology to calculate the correlation.

Given a joint multimodal signal $\mathcal{D}_n = \{\mathcal{D}_{nm}\}_{m=1}^M$, $K$-step adaptation of the context parameters $\{\phi_{nm}\}_{m=1}^M \in R^{S \times D_\phi}$ with MSFTs of $H$ multihead attention produces the attention score map $A_n$ with a shape of $K \times H \times MS \times MS$. We reshape this attention score map to $(K \times H \times S \times S) \times M \times M$, followed by reducing the leading dimensions to obtain an average attention score matrix $\bar{A}_n$ with a shape of $M \times M$. This matrix quantifies the average interactions among the learners, where the $(i, j)$ element of this matrix amounts to the average attention scores directed from the learner of modality $i$ towards the learner of modality $j$. Finally, we compute the Pearson correlation coefficient $C$ between these average attention score maps and the sizes of the support sets $\{|\mathcal{D}_{nm}|\}_m$ across $N$ sets of joint multimodal signals, where $C_{ij}$ amounts to the correlation between $\{\bar{A}_n(i,j)\}_n$ and $\{|\mathcal{D}_{nj}|\}_n$.

## F.2    EVOLVING ATTENTION MAPS THROUGH OPTIMIZATION STEPS

Figure 12 illustrates the evolving interactions among the learners through optimization steps $k = 1, \cdots, K$. We quantify this degree of interactions among the learners $\bar{A}^k \in \mathbb{R}^{M \times M}$ for $k = 1, \cdots, K$ by aggregating the attention scores of MSFTs over joint multimodal signals and optimization steps. This procedure involves the following steps: (1) We first reshape $A_n$ (defined as Section F.1) to $\mathbb{R}^{(H \times S \times S) \times K \times M \times M}$ (2) Then, this reshaped matrix is reduced to $\bar{A}_n^k \in \mathbb{R}^{K \times M \times M}$. (3) Finally, this reduced matrix is averaged across the set of N joint multimodal signals as $\bar{A}^k = \frac{1}{N} \sum_n \bar{A}_n^k$.

## F.3    ANALYSIS STUDY ON LEARNING TRAJECTORIES VIA MIA

We investigate how the learning trajectories of MIA and naive gradient descent (CAVIA) differ in situations where observations are sparse. To do this, we freeze the weight $\theta_m$ of the INR metatrained by CAVIA and re-metatrain it by adding the SFTs to the update process. After that, we visualized the learning trajectory of both methods according to the observation quality. Since the dimension of context parameters space is too high to visualize, we utilize Principal Component Analysis (PCA), following the methodology outlined in Li et al. (2018), to effectively project the trajectories into a two-dimensional space.

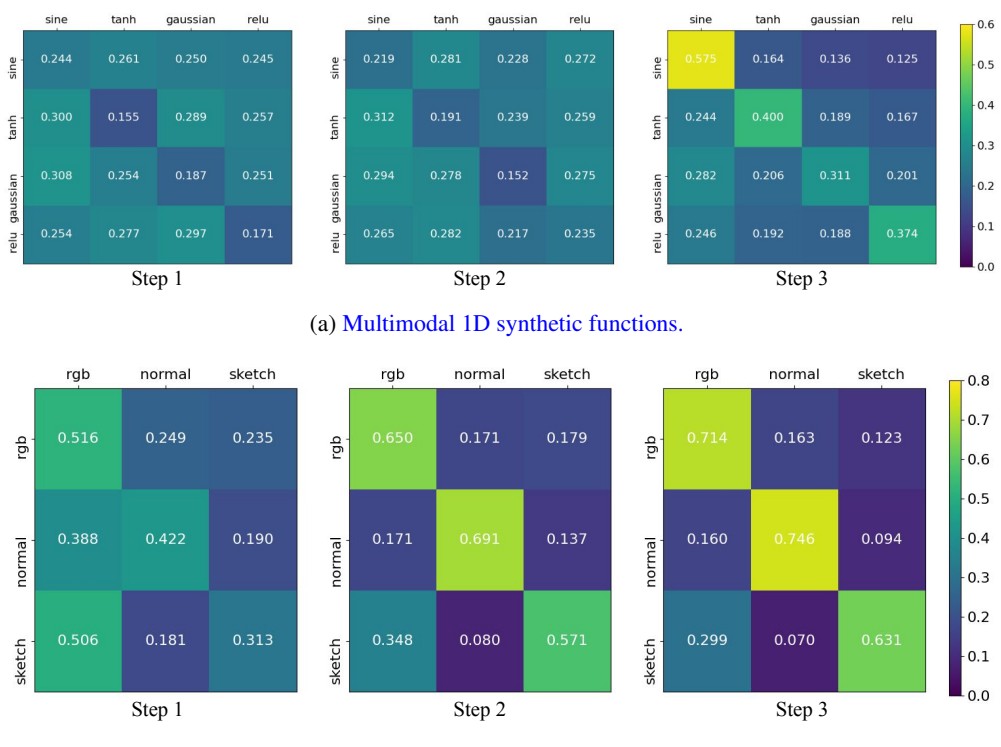

(a) Multimodal 1D synthetic functions.

(b) Multimodal 2D CelebA images.

Figure 12: Evolving interactions at each optimization step $k = 1, 2, 3$. Each $(i, j)$ element in the matrices indicates the average attention score assigned by the learner of modality $i$ to the learner of modality $j$. Please refer to Appendix F.2 for the detailed method for computing these matrices. The patterns show that learners tend to interact extensively with each other in the beginning (high off-diagonal attention scores in Step 1) and gradually attend more on themselves towards the end (high diagonal attention scores in Step 3). This highlights MIA's adaptability in ensuring a balanced utilization of multimodal interactions, emphasizing the necessity of applying multimodal adaptation iteratively for each optimization step.

Figure 13 illustrates that as the size of the support set for additional modalities expands, the learning trajectory with MIA shifts towards more effective solutions, moving away from local minima. This improvement is attributed to the SFTs' ability to enhance the noisy gradients derived from limited observations by utilizing information from other modalities' learners. Conversely, when adapting through standard gradient descent, the absence of cross-modal interactions results in a static learning trajectory, regardless of the observation from other modalities. In particular, when observations are scarce, this often results in poor performance compared to MIA, as it cannot escape to the local minima.

## F.4 ANALYSIS STUDY ON NEGATIVE TRANSFER FROM OTHER MODALITIES

In this study, we delve deeper into the impact of noisy state information from source modalities $m'$ on the target learner $m$. We evaluate the robustness of SFTs against potential negative transfer of incorrect information across modalities. To do this, we inject varying levels of Gaussian noise $\epsilon_{nm} \sim \mathcal{N}(0, \gamma I)$, into the gradient inputs $g_{nm'}^{(k)}$ of the SFTs. As the noise level $\gamma$ increases, the state information becomes irrelevant to the target learner. Specifically, we measure the average MSEs and the average attention scores between state tokens of the target learners, as varying the noise level from $10^{-5}$ to $1.0$.

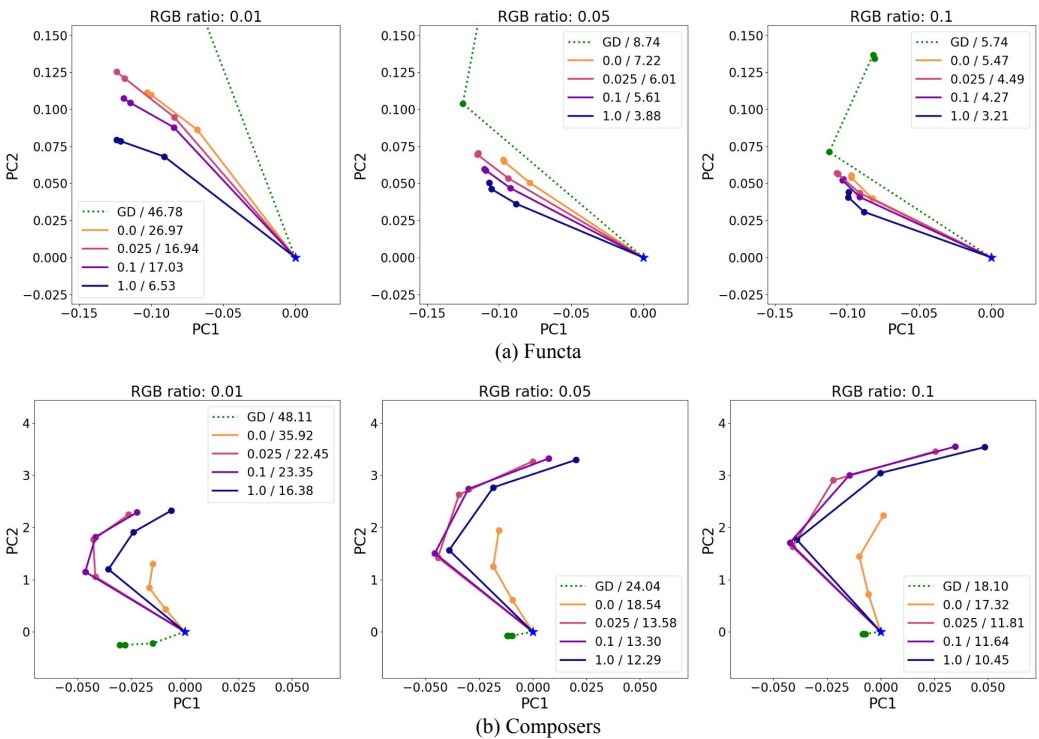

Figure 13: Learning trajectory of RGBs context parameters for a 3-step adaptation process with or without MIA, depending on the support set size of RGB. For each plot, the green dotted line represents the trajectory adapted with GD and the colored solid line represents the trajectory adapted with MIA according to the support set sizes of the different modalities (Normals + Sketches). The left side of the legend means the support set ratio of other modalities and the right side means the final reconstruction performance (MSEs).

Table 10: Results on CelebA dataset in MSEs ($\times 10^{-3}$) and averaged unimodal attention scores of MSFTs when injecting noise to gradients of other modalities' learner with varying noise level $\gamma$.

| | Functa | | | Composers | | |
|---|---|---|---|---|---|---|
| $\gamma$ | RGBs | Normals | Sketchs | RGBs | Normals | Sketchs |
| 0.00000 | 0.948 / 0.502 | 0.769 / 0.579 | 0.578 / 0.710 | 0.667 / 0.635 | 0.289 / 0.690 | 0.255 / 0.637 |
| 0.00001 | 0.976 / 0.525 | 0.774 / 0.644 | 0.581 / 0.720 | 0.664 / 0.630 | 0.288 / 0.696 | 0.255 / 0.643 |
| 0.00005 | 0.982 / 0.527 | 0.774 / 0.651 | 0.582 / 0.719 | 0.664 / 0.631 | 0.288 / 0.709 | 0.253 / 0.651 |
| 0.00010 | 0.987 / 0.527 | 0.775 / 0.653 | 0.582 / 0.718 | 0.673 / 0.656 | 0.290 / 0.727 | 0.253 / 0.658 |
| 0.00050 | 1.056 / 0.539 | 0.786 / 0.664 | 0.583 / 0.726 | 0.708 / 0.720 | 0.293 / 0.748 | 0.256 / 0.681 |
| 0.00100 | 1.092 / 0.549 | 0.796 / 0.677 | 0.584 / 0.741 | 0.714 / 0.724 | 0.293 / 0.752 | 0.256 / 0.687 |
| 0.00500 | 1.122 / 0.560 | 0.819 / 0.705 | 0.587 / 0.772 | 0.718 / 0.727 | 0.294 / 0.755 | 0.256 / 0.690 |
| 0.01000 | 1.124 / 0.561 | 0.823 / 0.709 | 0.588 / 0.775 | 0.717 / 0.727 | 0.294 / 0.755 | 0.256 / 0.690 |
| 0.10000 | 1.126 / 0.562 | 0.826 / 0.712 | 0.589 / 0.777 | 0.718 / 0.727 | 0.294 / 0.755 | 0.256 / 0.691 |
| 1.00000 | 1.127 / 0.562 | 0.826 / 0.713 | 0.589 / 0.778 | 0.718 / 0.727 | 0.294 / 0.755 | 0.256 / 0.691 |

As shown in Table 10, while there is a marginal rise in MSEs upon the introduction of noise, no additional performance degradation is noted beyond a certain noise level. Intriguingly, as the gradient noise from learners of other modalities increases, there is a concurrent increase in unimodal attention scores for the learner of the target modality. This strongly indicates that SFTs inherently have the ability to handle potential negative transfers across modalities.

F.5 ABLATION STUDY FOR COMPONENTS OF SFTs

Table 11: Ablation study on the components in SFTs on generalization and memorization ability. We report relative error reduction ($\uparrow$) achieved by ablated methods (2-8) over vanilla Composers (1) on multimodal 1D synthetic function dataset.

| | Modules | | | Synthetic | |
|---|---|---|---|---|---|
| | USFTs | MSFT | FusionMLPs | Generalization | Memorization |
| (1) | ✗ | ✗ | ✗ | 0.00 | 0.00 |
| (2) | ✗ | ✗ | ✓ | 38.9 | 54.2 |
| (3) | ✓ | ✗ | ✗ | 44.4 | 63.6 |
| (4) | ✓ | ✗ | ✓ | 43.1 | 68.0 |
| (5) | ✗ | ✓ | ✗ | 86.8 | 71.6 |
| (6) | ✗ | ✓ | ✓ | 86.8 | 76.7 |
| (7) | ✓ | ✓ | ✗ | 86.7 | 75.5 |
| (8) | ✓ | ✓ | ✓ | **88.7** | **81.6** |

This section provides an extensive ablation study encompassing all possible component combinations to investigate how each of them contribute to performance. We focus on the impact of each component on vanilla Composers in the synthetic dataset. The results are shown in Table 11.

The table shows the substantial impact of FusionMLPs in enhancing the weight updates of vanilla Composers when used independently (1 vs 2) or in combination with USFTs, MSFTs, or both (3 vs 4, 5 vs 6, 7 vs 8). This demonstrates the general versatility of FusionMLPs in enhancing gradient directions or magnitudes.

Moreover, we observe that incorporating USFTs significantly enhances memorization capabilities (1-2 vs 3-4). This emphasizes the advantageous role of USFTs in capturing modality-specific patterns within the learners' states.

In contrast, MSFTs excel in leveraging cross-modal interactions among the learners' states, leading to a substantial improvement in the generalization performances of Composers (3-4 vs 5-8).

Lastly, the most optimal performance is achieved when utilizing a combination of USFTs, MSFTs, and FusionMLPs, underlining the indispensable roles of each component within SFTs. This analysis validates the unique and crucial contribution of each component to the overall effectiveness of CAVIA.

