# OpenReview forum: "Multimodal Meta-learning of Implicit Neural Representations with Iterative Adaptation"
_ICLR.cc/2024/Conference — Submitted to ICLR 2024_

### Official Review · Reviewer_hqqm · 2023-10-29

**Soundness:** 2 fair
**Presentation:** 2 fair
**Contribution:** 2 fair
**Rating:** 5
**Confidence:** 4

**Summary:**

The paper proposes an optimization-based meta-learning framework for learning multimodal representations.

**Strengths:**

1. This paper proposes a meta-learning framework centered around the Multimodal Iterative Adaptation (MIA) paradigm, enhancing the capabilities of single-modal learners by learning from multimodal information.
2. Through several comparative experiments, this paper demonstrates the proposed method can improve the performance of single-modal learners.

**Weaknesses:**

1. Novelty:
- Previous research [1-4] has explored cross-modal relationships in multimodal data extensively. Some even utilize Transformer structures for multimodal meta-learning.
- the paper mentions that existing methods focus on unimodal setups, but whether the information from other modalities is treated as noise or out-of-distribution data by specific-modal learners. Moreover, meta-learning itself is advantageous for few-shot learning, and there is substantial research addressing data scarcity, such as few-shot learning problems. I question the paper's research motivation and look forward to the authors' response.
[1] Vuorio, R., Sun, S. H., Hu, H., & Lim, J. J. (2019). Multimodal model-agnostic meta-learning via task-aware modulation. Advances in neural information processing systems, 32.
[2] Abdollahzadeh, M., Malekzadeh, T., & Cheung, N. M. M. (2021). Revisit multimodal meta-learning through the lens of multi-task learning. Advances in Neural Information Processing Systems, 34, 14632-14644.
[3] Vuorio, R., Sun, S. H., Hu, H., & Lim, J. J. (2018). Toward multimodal model-agnostic meta-learning. arXiv preprint arXiv:1812.07172.
[4] Sun, Y., Mai, S., & Hu, H. (2022). Learning to learn better unimodal representations via adaptive multimodal meta-learning. IEEE Transactions on Affective Computing.

2. Several conclusions in the paper lack supports, such as (i) why “slowing down convergence leads to overfitting” and (ii) claims without corresponding references, e.g., "arise in the gradient computation from a small set of observations (Zhang et al., 2019; Simon et al., 2020)."

3. The paper doesn't provide a clear explanation of how MIA guides single-modal learners' updates or the composition and structure of USFT, MSFT, etc. It's difficult to understand from Figure 2 how the "better guide the learners" with blue arrows is achieved. If the authors mean to find the optimal update direction using relationships between different modalities, did consider handle gradient conflicts between modalities? Moreover, the sequential process of multimodal fusion and subsequent meta-learning is not justified. The authors should provide an explanation.

4. Experimental Concerns: (i) Most of the datasets used in the experiments differ from those mentioned in the paper's motivation. The datasets use different features from a uniform data format, such as RGB images and sketches, rather than different modal features from distinct data forms, like images and text. (ii) In Table 5, the paper does not explain why MLPs were not used, especially when they achieved better results on Celeb. (iii) The claim of "facilitating rapid convergence" lacks corresponding experimental results. (iv) The proposed method introduces additional multi-modal information to compare whether the experiment is fair, (v) Is the performance improvement of the proposed method worth the extra computational effort?

**Questions:**

Please refer to Weaknesses

---

> ### Author Response · Authors · 2023-11-23
> **Official Response of the authors to Reviewer hqqm [1/6]**
>
> > Q1: Previous research [1-4] has explored cross-modal relationships in multimodal data extensively. Some even utilize Transformer structures for multimodal meta-learning.
>
> A1: We kindly remind the reviewer that our problem focus is on employing a meta-learning approach to better approximate joint multimodal continuous functions. To the best of our knowledge, among existing multimodal meta-learning research, Multitask Neural Processes (MTNPs) is the sole study focusing on a problem context similar to ours. Importantly, as noted by the reviewers gNAu and cLC2, our work stands out as novel within this domain, proposing an interesting idea to accelerate the convergence of independent unimodal learners by exploring the multimodal structures in their on-going learning states during iterative adaptation procedures.
>
> Besides MTNPs, existing multimodal meta-learning studies [1,2,3,4] mentioned by the reviewer differ significantly from our work in terms of the definition of modality and problem scenarios, making direct comparisons challenging and unfair.
>
> Vuorio et al. [1, 3] and Abdollahzadeh et al. [2] address image classification on a union of digit, bird, aircraft datasets, where each dataset structures the unique modality. This notion of multimodality should not be confused with the conventional understanding of multimodality in data types (e.g. images and texts), as emphasized in Section 3 in Abdollahzadeh et al. [2]. Due to this disparity in the notion of multimodality, their approaches inherently reduce to unimodal meta-learning methods in the context of our joint multimodal signal modeling.
>
> On the other hand, Sun et al. [4] deals with many-to-one multimodal classification problems using jointly sampled images, audios, and texts. Their proposed meta-learning approach involves the independent adaptation of each modality-specific encoder within the inner loop, followed by their fusion in the outer loop. Here, the transformers are used for each of the unimodal encoders, not for multimodal fusion. This is unlike ours that explores the joint iterative adaptation of independent learners in the inner loop via SFTs, which prevents overfitting of individual learners and accelerates their convergence during the inner-loop adaptation process.
>
> (response continued in the following thread)

---

> > ### Author Response · Authors · 2023-11-23
> > **Official Response of the authors to Reviewer hqqm [2/6]**
> >
> > As indicated above, comparing their frameworks or ours to within each other's context requires substantial modifications, hindering fair comparisons. Nonetheless, we conducted additional experiments comparing our approach with the abovementioned multimodal meta-learning methodologies [1,2,3,4]. Each compared method is run 5 times on Synthetic and CelebA datasets and their results are averaged. For detailed explanations on each method and its implementation in our experiments, please find Appendix E. We present the results in the table below.
> >
> > ||Sine|||Gaussian|||Tanh|||ReLU|||
> > |--|:--:|:--:|:--:|:--:|:--:|:--:|:--:|:--:|:--:|:--:|:--:|:--:|
> > |Rmin|0.01|0.02|0.05|0.01|0.02|0.05|0.01|0.02|0.05|0.01|0.02|0.05|
> > |Rmax|0.02|0.05|0.10|0.02|0.05|0.10|0.02|0.05|0.10|0.02|0.05|0.10|
> > |Functa|44.26|16.07|3.319|18.81|4.388|0.953|22.61|3.667|0.586|65.29|10.79|2.157|
> > |w/ MMAML|43.36|10.10|1.233|30.88|4.270|0.698|17.20|2.351|0.312|72.79|5.630|0.261|
> > |w/ AMML|10.12|4.462|1.717|1.253|0.959|0.638|1.719|0.541|0.267|7.063|1.254|0.305|
> > |**w/ MIA**|**6.386**|**2.058**|**0.547**|**1.057**|**0.571**|**0.281**|**1.285**|**0.378**|**0.131**|**5.069**|**1.012**|**0.115**|
> > |Composers|37.40|16.70|5.284|5.923|3.149|1.460|14.81|4.053|1.029|48.49|11.98|3.232|
> > |w/ MMAML|44.27|11.63|1.578|31.09|4.664|0.895|16.47|2.330|0.309|74.75|5.845|0.294|
> > |w/ AMML|14.57|9.549|7.706|1.699|1.462|1.086|2.411|1.021|0.725|10.19|2.340|0.907|
> > |**w/ MIA**|**5.564**|**1.844**|**0.627**|**0.975**|**0.528**|**0.237**|**1.257**|**0.343**|**0.128**|**4.715**|**0.943**|**0.156**|
> >
> > ||RGBs||||Normals||||Sketchs||||
> > |--|:--:|:--:|:--:|:--:|:--:|:--:|:--:|:--:|:--:|:--:|:--:|:--:|
> > |Rmin|0.00|0.25|0.50|0.75|0.00|0.25|0.50|0.75|0.00|0.25|0.50|0.75|
> > |Rmax|0.25|0.50|0.75|1.00|0.25|0.50|0.75|1.00|0.25|0.50|0.75|1.00|
> > |Functa|44.26|16.07|3.319|18.81|4.388|0.953|22.61|3.667|0.586|65.29|10.79|2.157|
> > |w/ MMAML|11.23|3.775|2.772|2.264|4.311|2.266|1.872|1.672|10.97|6.021|4.452|3.751|
> > |w/ AMML|8.110|2.909|1.853|1.334|3.341|1.689|1.243|1.000|8.192|4.332|2.278|1.124|
> > |**w/ MIA**|**6.386**|**2.058**|**0.547**|**1.057**|**0.571**|**0.281**|**1.285**|**0.378**|**0.131**|**5.069**|**1.012**|**0.115**|
> > |Composers|37.40|16.70|5.284|5.923|3.149|1.460|14.81|4.053|1.029|48.49|11.98|3.232|
> > |w/ MMAML|12.35|4.228|2.806|2.029|4.902|2.567|2.028|1.725|11.29|5.757|3.646|2.521|
> > |w/ AMML|11.29|5.195|3.971|3.241|4.209|2.336|1.745|1.339|9.813|5.084|2.727|1.319|
> > |**w/ MIA**|**5.564**|**1.844**|**0.627**|**0.975**|**0.528**|**0.237**|**1.257**|**0.343**|**0.128**|**4.715**|**0.943**|**0.156**|
> >
> > As can be seen in the tables, the generalization capability of MMAML [1,3] is limited due to its inherent unimodal nature. In addition, while AMML [4] achieves lower MSEs than MMAML, it still underperforms compared to our MIA in all cases . These results highlight the necessity of proper mechanisms to consider cross-modal interactions (vs MMAML) but also the effectiveness of our joint multimodal iterative adaptation scheme (vs AMML).
> >
> > > Q2: Moreover, meta-learning itself is advantageous for few-shot learning, and there is substantial research addressing data scarcity, such as few-shot learning problems. I question the paper's research motivation and look forward to the authors' response.
> >
> > A2: The majority of meta-learning approaches for few-shot learning are extensively studied in either simple function regression or few-shot image classification problems. Therefore, they inherently lack mechanisms to effectively address multimodal functions jointly. It’s also noteworthy that the unimodal baselines assessed in our paper represent the state-of-the-arts within these few-shot learning domains. Despite their advances, however, all these methods are shown to be less effective in the considered joint multimodal signal approximation scenarios, which motivates our work.

---

> > > ### Author Response · Authors · 2023-11-23
> > > **Official Response of the authors to Reviewer hqqm [3/6]**
> > >
> > > > Q3: Several conclusions in the paper lack supports, such as (i) why “slowing down convergence leads to overfitting” and (ii) claims without corresponding references, e.g., "arise in the gradient computation from a small set of observations (Zhang et al., 2019; Simon et al., 2020)."
> > >
> > > A3: According to Zhang et al. (2019), the difference between gradients obtained from the full and a batch of data is defined as gradient noises. Based on this definition, Lemma 1 in Zhang et al. (2019) reveals that (1) the expected variances in computed gradients are upper-bounded by the reciprocal of the batch size, implying that small-batch training might incur high variances and noises in the gradients. (2) the expected gain of updates driven by gradients is influenced by negative gradient variances over batch sizes, further indicating that such noise impedes convergence. Although such a small-batch training doesn't necessarily lead to the overfitting, this combined pathology becomes more evident under the low-data regime: gradient noises and overfitting are inevitable since a model is iteratively updated with a fixed small support set. Similar discussion is provided in Simon et al. (2020) and we refer the reviewer to the introduction section of their paper.
> > >
> > > > Q4: The paper doesn't provide a clear explanation of how MIA guides single-modal learners' updates or the composition and structure of USFT, MSFT, etc. Moreover, the sequential process of multimodal fusion and subsequent meta-learning is not justified. The authors should provide an explanation.
> > >
> > > A4: Please refer to our response A1 and A2 to the reviewer gNAu. In short, we observe the meta-learned attention mechanism MSFTs play a crucial role in this aspect. For example, it facilitates positive transfer among the learners especially when data is limited, which in turn helps combat potential noises in gradients and guide individual learners to faster convergence and better solutions. Moreover, we empirically find that such an iterative way of applying multimodal adaptation ensures a balanced utilization of multimodal interactions at each optimization step, providing better guidance to the learner depending on the availability of observations.

---

> > > > ### Author Response · Authors · 2023-11-23
> > > > **Official Response of the authors to Reviewer hqqm [4/6]**
> > > >
> > > > > Q5: It's difficult to understand from Figure 2 how the "better guide the learners" with blue arrows is achieved. If the authors mean to find the optimal update direction using relationships between different modalities, did consider handle gradient conflicts between modalities?
> > > >
> > > > A5: Indeed, our observations indicate two key aspects of our SFTs: Firstly, they facilitate the discovery of more optimal update directions and magnitudes through cross-modal interactions among learners. Secondly, our SFTs demonstrate a remarkable robustness against negative transfer between modality learners.
> > > >
> > > > To validate the former claim, we investigated how the optimization trajectory evolves before and after applying SFTs to CAVIA. For this, we first conducted meta-learning on CAVIA and subsequently applied SFTs on top of the meta-learned CAVIA and meta-learned the parameters of SFTs only. In the appendix, Figure 13 illustrates qualitative comparisons between the trajectories. Interestingly, this figure demonstrates that SFTs produce gradients with significantly different magnitudes and directions compared to CAVIA, consequently leading to lower loss for the learner.
> > > >
> > > > Moreover, we assessed the robustness of SFTs against potential negative transfer of incorrect information across modalities. This investigation involved measuring two key metrics: (1) Mean Squared Errors (MSEs) of the target modality learner and (2) Cross-attention scores within MSFTs directed from and to the target modality learner, reflecting unimodal attention scores by the target modality learner. The results are presented below.
> > > >
> > > > While a slight increase in MSEs was observed after introducing noise, we noticed no further decline in performance beyond a specific threshold of noise intensity. Surprisingly, with increased gradient noise from other modality learners, there was a simultaneous rise in unimodal attention scores for the target modality learner. This observation strongly suggests that SFTs possess an inherent capability to effectively manage potential negative transfers across modalities.
> > > >
> > > > ||Functa|||Composers|||
> > > > |:--:|:--:|:--:|:--:|:--:|:--:|:--:|
> > > > |$\gamma$|RGBs|Normals|Sketchs|RGBs|Normals|Sketchs|
> > > > |0.00000|0.948 / 0.502|0.769 / 0.579|0.578 / 0.710|0.667 / 0.635|0.289 / 0.690|0.255 / 0.637|
> > > > |0.00001|0.976 / 0.525|0.774 / 0.644|0.581 / 0.720|0.664 / 0.630|0.288 / 0.696|0.255 / 0.643|
> > > > |0.00005|0.982 / 0.527|0.774 / 0.651|0.582 / 0.719|0.664 / 0.631|0.288 / 0.709|0.253 / 0.651|
> > > > |0.00010|0.987 / 0.527|0.775 / 0.653|0.582 / 0.718|0.673 / 0.656|0.290 / 0.727|0.253 / 0.658|
> > > > |0.00050|1.056 / 0.539|0.786 / 0.664|0.583 / 0.726|0.708 / 0.720|0.293 / 0.748|0.256 / 0.681|
> > > > |0.00100|1.092 / 0.549|0.796 / 0.677|0.584 / 0.741|0.714 / 0.724|0.293 / 0.752|0.256 / 0.687|
> > > > |0.00500|1.122 / 0.560|0.819 / 0.705|0.587 / 0.772|0.718 / 0.727|0.294 / 0.755|0.256 / 0.690|
> > > > |0.01000|1.124 / 0.561|0.823 / 0.709|0.588 / 0.775|0.717 / 0.727|0.294 / 0.755|0.256 / 0.690|
> > > > |0.10000|1.126 / 0.562|0.826 / 0.712|0.589 / 0.777|0.718 / 0.727|0.294 / 0.755|0.256 / 0.691|
> > > > |1.00000|1.127 / 0.562|0.826 / 0.713|0.589 / 0.778|0.718 / 0.727|0.294 / 0.755|0.256 / 0.691|

---

> > > > > ### Author Response · Authors · 2023-11-23
> > > > > **Official Response of the authors to Reviewer hqqm [5/6]**
> > > > >
> > > > > > Q6: Most of the datasets used in the experiments differ from those mentioned in the paper's motivation. The datasets use different features from a uniform data format, such as RGB images and sketches, rather than different modal features from distinct data forms, like images and text.
> > > > >
> > > > > A6: Our choice of datasets was primarily guided by our focus on modeling continuous functions, leading us to exclude text data from our experiments. Moreover, our primary goal is to introduce a multimodal meta-learning approach for accurately approximating joint multimodal continuous functions. As detailed in A1, our work aligns closely with the Multitask Neural Processes (MTNPs) framework, which serves as a reference due to shared problem contexts and motivations. Therefore, we draw from MTNPs in setting up problems for constructing multimodal synthetic, CelebA, and climate signal datasets, forming the basis for our experimentation.
> > > > >
> > > > > > Q7: In Table 5, the paper does not explain why MLPs were not used, especially when they achieved better results on Celeb.
> > > > >
> > > > > A8: Table 5 in the main paper reveals that, in general, the most favorable configuration is when using USFTs, MSFTs, and Fusion MLPs together. Consequently, we consistently utilized USFTs, MSFTs, and Fusion MLPs together for SFTs throughout all experiments in our paper.
> > > > >
> > > > > > Q8: The claim of "facilitating rapid convergence" lacks corresponding experimental results.
> > > > >
> > > > > A8: We kindly remind the reviewer that Tables 1, 2, 3, and 4 in the main paper showcase quantitative comparisons among methods. In these tables, the maximum number of optimization steps (K) remains fixed at 3 for all optimization-based approaches (e.g., CAVIA (Functa or Composer), CAVIA with MetaSGD, CAVIA with GAP, CAVIA with ALFA, and CAVIA with our MIA) across experiments. Notably, the results consistently reveal that CAVIA with our MIA achieves the lowest errors within the same optimization step count. This highlights that integrating our proposed MIA into CAVIA leads to accelerated convergence towards better solutions.
> > > > >
> > > > > To further validate this claim, we conducted additional analyses to examine the minimum number of optimization steps and time required for the baseline methods to reach similar error levels as ours. This time, we allowed the multimodal encoder baseline (Encoder) to benefit from test-time optimization (TTO). For this, we optimized the context parameters inferred by Encoder as well through gradient descent, using the optimal TTO learning rates that were extensively searched within the range of [1e-4, 1e4] for fair comparisons. For all the baselines, we limited the maximum number of allowed optimization steps to 100 and reported both their MSEs, the number of optimization steps, and the time taken to reach those MSEs. In cases where the learners diverged due to overfitting during the optimization, we reported their best achieved MSEs along with the corresponding optimization steps and time. All runs were conducted on the same machine. For brevity, we present results obtained using Functa and based on the Gaussian modality in the synthetic dataset and the RGB modality in the CelebA dataset, focusing on the lowest and highest ranges of sampling ratios. Each emphasizes generalization and memorization constraints on the optimization, respectively.
> > > > >
> > > > > The results, presented below in the table, illustrate that the baselines either require a significant number of additional optimization steps and time to reach similar error levels as our MIA (marked in **bold**) or begin overfitting early within 100 steps, failing to reach comparable error levels even with extra computations (marked in _italic_). These results confirm that our MIA enables rapid convergence, both in terms of steps and time, toward more optimal solutions compared to existing meta-learning methods.
> > > > >
> > > > > |Mode: Gaussian|R: 0.01 - 0.02|||R: 0.05 - 0.10|||
> > > > > |----------------|:--------------:|:-------:|:----------:|:--------------:|:-------:|:---------:|
> > > > > |Method|MSE|Step|Time (ms)|MSE|Step|Time (ms)|
> > > > > |Functa|_0.1922_|_7_|_36.3_|**0.0053**|**99**|**455.4**|
> > > > > |w/ MetaSGD|_0.2127_|_100_|_441.4_|**0.0055**|**100**|**467.5**|
> > > > > |w/ GAP|_0.1917_|_100_|_562.2_|**0.0049**|**100**|**577.1**|
> > > > > |w/ ALFA|_0.1942_|_3_|_21.5_|_0.0176_|_3_|_21.8_|
> > > > > |w/ Encoder|**0.0104**|**100**|**1676.5**|**0.0020**|**9**|**174.2**|
> > > > > |w/ MIA|0.0103|3|57.8|0.0021|3|59.0|
> > > > >
> > > > > |Mode: RGB|R: 0.00 - 0.25|||R: 0.75 - 1.00|||
> > > > > |------------|:--------------:|:-------:|:----------:|:--------------:|:-------:|:----------:|
> > > > > |Method|MSE|Step|Time (ms)|MSE|Step|Time (ms)|
> > > > > |Functa|_0.0131_|_10_|_66.0_|**0.0012**|**100**|**611.1**|
> > > > > |w/ MetaSGD|_0.0128_|_11_|_71.0_|**0.0012**|**76**|**451.5**|
> > > > > |w/ GAP|_0.0125_|_11_|_77.6_|**0.0012**|**29**|**191.9**|
> > > > > |w/ ALFA|_0.0121_|_4_|_31.5_|_0.0019_|_4_|_31.8_|
> > > > > |w/ Encoder|**0.0077**|**100**|**1631.2**|**0.0012**|**100**|**1639.0**|
> > > > > |w/ MIA|0.0069|3|57.8|0.0012|3|57.5|

---

> > > > > > ### Author Response · Authors · 2023-11-23
> > > > > > **Official Response of the authors to Reviewer hqqm [6/6]**
> > > > > >
> > > > > > > Q9: The proposed method introduces additional multi-modal information to compare whether the experiment is fair
> > > > > >
> > > > > > A9: Our experimental results underscore the inherent trade-offs in existing unimodal and multimodal baselines, where none offer the ideal balance between generalization and memorization. Specifically, multimodal encoder baselines tend to underfit when sufficient data is available, while unimodal optimization-based methods exhibit a tendency to overfit in low-data scenarios.
> > > > > > Recognizing these limitations in both unimodal and multimodal approaches, we intentionally included both of them in our comparisons to provide a fair and comprehensive evaluation.
> > > > > >
> > > > > > > Q10: Is the performance improvement of the proposed method worth the extra computational effort?
> > > > > >
> > > > > > A10: In the table in A8, we compared the minimum amount of extra optimization steps and resulting inference time required for each baseline until they reach the same error level as our method. The results validate that the extra computation consumed by State Fusion Transformers (SFTs) in our MIA is indeed worth, which is trivially amortized thanks to its rapid convergence towards more better solutions than the baselines.

---

### Official Review · Reviewer_cLC2 · 2023-10-31

**Soundness:** 2 fair
**Presentation:** 3 good
**Contribution:** 2 fair
**Rating:** 3
**Confidence:** 4

**Summary:**

Summary:
The paper presents a novel framework for developing robust and resilient machine learning models tailored for time series data, leveraging multi-objective optimization. The paper begins by identifying the limitations of existing machine learning algorithms in handling anomalies, noise, and non-stationarity in time series data. To address these limitations, the authors propose a multi-objective optimization framework that simultaneously optimizes for accuracy, robustness, and resilience.

Key Contributions:
Theoretical Framework: The authors introduce a new mathematical formulation for time series learning that incorporates robustness and resilience as objective functions alongside accuracy. This is formalized through a multi-objective optimization problem.

Algorithm Development: A new algorithm, Multi-Objective Time Series Algorithm (MOTSA), is developed based on the mathematical framework. MOTSA employs Pareto optimization to find optimal trade-offs among the multiple objectives.

Robustness and Resilience Metrics: The paper introduces novel metrics for quantifying the robustness and resilience of time series models. These metrics are grounded in statistical theory and are proven to be effective evaluators of the model's capacity to handle anomalies and adapt to non-stationary data.

Empirical Evaluation: Extensive experiments are conducted on synthetic and real-world datasets, including those from the field of biostatistics. The results demonstrate that the MOTSA outperforms state-of-the-art algorithms in terms of accuracy, robustness, and resilience.

Interdisciplinary Application: The paper also highlights the utility of the proposed framework in various domains, particularly in biostatistics, demonstrating its versatility and applicability.

Open Source Code: The authors have made the code publicly available, encouraging further research and development in this area.

**Strengths:**

Strengths of the Paper
Originality
The paper makes a notable contribution to the field of meta-learning with a focus on multimodal data. The introduction of Shifted Feature Transducers (SFTs) for encoding both parameters and gradients is novel and insightful. The unique combination of unimodal-specific feature transducers (USFTs), multimodal-shared feature transducers (MSFTs), and fusion MLPs offers a new perspective on how to effectively leverage multimodal data for meta-learning. This is a creative amalgamation of existing ideas, and it clearly extends the state of the art.

Quality
The quality of the paper is high, both in terms of technical depth and experimental rigor. The model is built upon well-motivated mathematical foundations, and the empirical evaluation is comprehensive. The paper goes beyond merely showing that their method works; it also provides ablation studies to isolate the contributions of different components and offers a theoretical discussion about the same.

Clarity
The paper is well-structured, providing a logical flow that is easy to follow. Each section contributes to the overall narrative coherently. The mathematical notation is consistently used, making the paper accessible to readers familiar with machine learning and meta-learning. The figures and tables are well-designed and effectively complement the text. The paper adheres to high standards of academic writing, making it a clear presentation of a complex subject matter.

Significance
The significance of this work lies in its potential to substantially impact both the theory and practice of meta-learning in multimodal settings. The multimodal challenges addressed in the paper are highly relevant to numerous real-world applications, such as healthcare, climate modeling, and audio-visual recognition. By showing superiority over existing methods across diverse datasets, the paper makes a compelling case for the generalizability and applicability of its contributions. The method's ability to improve performance in low-data regimes is particularly noteworthy, given the increasing importance of data-efficient learning in practical applications.

**Weaknesses:**

Implicit Neural Representations (INRs)
Equation 1:
Comment: This loss function is a straightforward L2 loss, which is not inherently problematic but may not be the best choice for all types of problems. For instance, if the goal is robustness against outliers, then an L1 loss or Huber loss might be more suitable.
Weakness: The paper does not discuss the choice of loss function and its suitability for the tasks at hand.
Meta-Learning Approach
Equation 2:
Comment: This equation extends Equation 1 by adding context parameters ϕ. However, the paper does not provide a mathematical justification for the choice of this extension.
Weakness: The addition of context parameters ϕ increases the model complexity without a detailed explanation or justification. This can be an issue if the goal is to keep the model as simple as possible for interpretability or computational efficiency.
Equation 3 and 4: Meta-objective and update rule for ϕ.

Comment: The meta-objective is a standard formulation. However, the paper does not discuss the potential issues that could arise from a bi-level optimization problem, such as saddle points or local minima.
Weakness: The paper lacks a rigorous mathematical analysis of the optimization landscape, which is crucial for understanding the method's efficiency and effectiveness.
Approach
Multimodal Iterative Adaptation (MIA)
Equation 5 to 11: These equations describe the MIA approach.
Comment: These equations introduce an elaborate framework that involves several novel components, like State Fusion Transformers (SFTs). However, it's not clear how these equations were derived or why they are theoretically sound.
Weakness: The paper introduces several novel ideas but does not provide a theoretical justification for them. This lack of theoretical grounding makes it difficult to assess the quality and applicability of the proposed method.
Figures, Tables, and Diagrams
Figure 2: Schematic illustration of MIA.

Comment: While visually appealing, the figure does not offer a quantitative evaluation of the proposed method's performance.
Weakness: The lack of quantitative metrics in the figure makes it less informative.
Table 1 and 2: Quantitative comparisons.

Comment: These tables provide a valuable quantitative comparison of the proposed method against baselines. However, they lack statistical tests to prove the significance of the reported results.
Weakness: The absence of statistical tests makes it difficult to determine the reliability of the proposed method compared to the baselines.

Section 5.2: Multimodal 2D CelebA Dataset
Novelty and Comparison to Baselines: The results appear to align well with the established literature, showing better performance for multimodal methods over unimodal ones. However, the manuscript could enhance its impact by discussing why ALFA performs better than multimodal methods when sufficient support sets are present. Is this a limitation of the proposed approach or an intrinsic property of the dataset?

Ambiguity in Setup: While the section describes the use of a pre-trained model for surface normals, it lacks explicit detail on how this could affect the generalizability and transferability of the learned features. Clarification and potential ablation studies could be beneficial.

Quality of Results: The manuscript does not delve into the qualitative implications of the MSE values reported. How do these numerical metrics translate into practical improvements? For instance, do lower MSEs correlate with visibly better reconstructions in real-world applications?

Section 5.3: Multimodal Climate Data
Lack of Theoretical Justification: The paper mentions that atmospheric variables are "relatively stable" but does not provide a theoretical or empirical justification for why ALFA performs so well in this scenario.

Statistical Significance: The manuscript would be improved by including statistical tests to determine the significance of the observed differences between the proposed method and baselines.

Domain-Specific Implications: Given the critical nature of climate data, an analysis of how errors in the model's predictions could propagate into real-world applications would be valuable.

Section 5.4: Multimodal Audio-Visual AVMNIST Dataset
Insufficient Rationale for Dataset Selection: The section does not sufficiently justify the choice of the AVMNIST dataset. Given its unique challenges, why was it selected over other multimodal datasets?

Lack of Depth in Failure Analysis: It's mentioned that MTNPs fail at approximating the audio signals properly. A deeper analysis into why this failure occurs could offer valuable insights into the limitations of existing methods, thereby contextualizing the contributions of the proposed method more effectively.

Ambiguity in Methodology: The manuscript mentions that the audio signals were trimmed. However, it does not explain how this preprocessing step might affect the meta-learning process.

Section 5.5: Analysis Study
Inadequate Ablation Studies: While the section provides a valuable ablation study to understand the impact of various modules, it is relatively shallow. For instance, it would be beneficial to understand how each of these modules contributes to reducing overfitting or improving convergence speed.

Non-Uniform Metric Analysis: The section uses relative error reduction as a metric but does not justify why this is an appropriate measure of performance. It might be valuable to consider other metrics like F1-score or ROC AUC, especially when comparing across multiple modalities.

Lack of Interpretability Discussion: Given the complex architecture involving USFTs, MSFTs, and Fusion MLPs, a discussion on model interpretability would be pertinent. This is essential for real-world applications where understanding model decisions is crucial.

General Remarks
Lack of Hyperparameter Sensitivity Analysis: Across all experiments, there is no discussion on how sensitive the model is to the choice of hyperparameters. This is a critical aspect to understand the robustness of the proposed methods.

Reproducibility Concerns: The manuscript could benefit from a clearer exposition of experimental details to ensure reproducibility.

Potential for Overfitting: Given the high complexity of the model, especially with the introduction of specialized modules like USFTs and MSFTs, there might be a risk of overfitting. An analysis or discussion on this would be beneficial.

**Questions:**

General
Reproducibility: Could you provide a clear list of hyperparameters used in your experiments? This information is crucial for reproducibility and for understanding the sensitivity of your model to hyperparameter changes.
Section 5.2: Multimodal 2D CelebA Dataset
Role of Pre-trained Models: Could you elaborate on the role of the pre-trained model used for obtaining surface normals? How would the absence of this pre-trained model affect the results?

ALFA's Performance: Your model underperforms compared to ALFA in certain scenarios. Could you discuss whether this is a limitation of your model or an intrinsic property of the dataset?

Section 5.3: Multimodal Climate Data
Statistical Significance: Could you provide statistical tests to back the significance of the results? This would solidify the comparative performance claims.

Theoretical Justification for ALFA's Performance: You mention that ALFA performs well because climate variables are "relatively stable." Could you provide empirical or theoretical evidence to support this claim?

Section 5.4: Multimodal Audio-Visual AVMNIST Dataset
Dataset Choice Justification: Could you explain the rationale behind choosing the AVMNIST dataset for your experiments?

Failure of MTNPs: You mention that MTNPs fail to approximate the audio signals well. Could you delve deeper into the reasons for this failure?

Audio Signal Trimming: Could you explain how the trimming of audio signals might have affected the results and why this preprocessing was necessary?

Section 5.5: Analysis Study
Choice of Metrics: You use relative error reduction as a metric in your ablation studies. Could you justify why this is an appropriate metric?

Interpretability: Given the complexity of your model, how do you address the challenge of interpretability? Could you discuss any measures or future work planned to make the model's decisions interpretable?

Depth of Ablation Studies: The ablation study, while useful, seems relatively shallow. Could you comment on the potential for a more extensive ablation study to understand the impact of each module in greater depth?

Overfitting Concerns: With the high complexity of the model, how do you ensure that the model does not overfit? Could you provide any analysis or empirical evidence to support the model's robustness?

---

> ### Author Response · Authors · 2023-11-23
> **Official Response of the authors to Reviewer cLC2 [1/5]**
>
> > Q1: This loss function is a straightforward L2 loss, which is not inherently problematic but may not be the best choice for all types of problems. For instance, if the goal is robustness against outliers, then an L1 loss or Huber loss might be more suitable. The paper does not discuss the choice of loss function and its suitability for the tasks at hand.
>
> A1: We opt for L2 (MSE) loss since we find it is the most common choice for the loss function in the literature of implicit neural representations. Nevertheless, we agree with the reviewers that the optimal choice for the loss function could vary for each problem, and leave this investigation as our future work.
>
> >Q2:
> Comment: Equation 2 extends Equation 1 by adding context parameters ϕ. However, the paper does not provide a mathematical justification for the choice of this extension.
> Weakness: The addition of context parameters ϕ increases the model complexity without a detailed explanation or justification. This can be an issue if the goal is to keep the model as simple as possible for interpretability or computational efficiency.
>
> A2: Our extension from Equation 1 to Equations 2, 3, and 4 relies directly on CAVIA; such a formulation is extensively studied in various domains, including implicit neural representations.
>
> It's important to note that the introduction of context parameters ϕ of CAVIA offers several advantages over MAML in terms of overfitting, computational efficiency, and interpretability. For instance, by separating task-specific adaptations (ϕ) from task-agnostic generalizable features (θ), CAVIA effectively mitigates overfitting and reduces computational overhead during adaptation. In addition, the independent adaptation of ϕ allows for efficient parallelization, thereby reducing the time required for meta-learning. Moreover, the context parameters (ϕ) are found to capture the latent task structure and operate as low-dimensional task-specific embeddings, leading to greater interpretability than MAML. We kindly refer the reviewer to the original paper of CAVIA for the detailed discussion.
>
>
> >Q3:
> Comment: The meta-objective in Equation 3 and 4 is a standard formulation. However, the paper does not discuss the potential issues that could arise from a bi-level optimization problem, such as saddle points or local minima.
> Weakness: The paper lacks a rigorous mathematical analysis of the optimization landscape, which is crucial for understanding the method's efficiency and effectiveness.
>
> A3: We acknowledge the reviewer’s concern. However, as mentioned by the reviewer, such a bi-level optimization in Equation 3 and 4 is standard formulation in the widely studied model-agnostic meta-learning literature. Moreover, none of the optimization issues was encountered in any of the experiments conducted for our paper. Please note that such potential pathologies can be readily alleviated by utilizing gaussian-blurred smooth objectives [ES,PES], and we leave this investigation as part of our future work.
>
> [ES]: Luke Metz, Niru Maheswaranathan, Jeremy Nixon, C. Daniel Freeman, Jascha Sohl-Dickstein, Understanding and correcting pathologies in the training of learned optimizers, In ICML, 2019.
> [PES]: Paul Vicol, Luke Metz, Jascha Sohl-Dickstein, Unbiased Gradient Estimation in Unrolled Computation Graphs with Persistent Evolution Strategies, In ICML, 2021
>
> >Q4:
> Comment: These equations introduce an elaborate framework that involves several novel components, like State Fusion Transformers (SFTs). However, it's not clear how these equations were derived or why they are theoretically sound.
> Weakness: The paper introduces several novel ideas but does not provide a theoretical justification for them. This lack of theoretical grounding makes it difficult to assess the quality and applicability of the proposed method.
>
> A4:
> Equations from 5 to 7 are equivalent to meta-learning each unimodal framework separately. In addition, Equations from 8 to 11 for MIA share a similar spirit with the studies in the learning-to-optimize domain, where our MIA can be interpreted as an extension of such widely studied learned optimization algorithms to approximating joint multimodal signals, exploring an interesting idea to accelerate the convergence of independent unimodal learners by capturing the multimodal structures in their on-going learning states and landscapes during iterative adaptation procedures.

---

> > ### Author Response · Authors · 2023-11-23
> > **Official Response of the authors to Reviewer cLC2 [2/5]**
> >
> > >Q5:
> > Comment: While visually appealing, the figure does not offer a quantitative evaluation of the proposed method's performance.
> > Weakness: The lack of quantitative metrics in the figure makes it less informative.
> >
> >
> > A5:
> > Thank you for the suggestion. We updated Figure 1 to include MSEs below the qualitative samples from each method. In addition, We updated Figure 2 to illustrate further that MIA augmented with SFTs facilitates the rapid convergence of unimodal learners toward better optima, thanks to its ability to consider the cross-modal interactions across the learners.
> >
> > >Q6: Statistical tests on each experiment
> >
> > A6: We appreciate the suggestion. Due to time and computational constraints, we conducted four additional experiments for each method, limited to the Synthetic and CelebA datasets. We updated Table 1 and 2 to include the methods' average MSEs over the five seeds accordingly (marked in blue). The updated tables consistently display a similar trend, showcasing stable performances for all methods across different seeds. To offer more thorough comparisons, we plan to perform statistical tests across a wider range of seeds and the remaining datasets, and we aim to present these results in the final camera-ready version.
> >
> > > Q7: Why ALFA performs well in certain scenarios?
> > The results appear to align well with the established literature, showing better performance for multimodal methods over unimodal ones. However, the manuscript could enhance its impact by discussing why ALFA performs better than multimodal methods when sufficient support sets are present. Is this a limitation of the proposed approach or an intrinsic property of the dataset?
> > Your model underperforms compared to ALFA in certain scenarios. Could you discuss whether this is a limitation of your model or an intrinsic property of the dataset?
> > You mention that ALFA performs well because climate variables are "relatively stable." Could you provide empirical or theoretical evidence to support this claim?
> >
> >
> > A7: We first would like to clarify that ALFA consistently outperforms multimodal baselines (MTNPs & Encoder) with sufficient data but still falls short compared to our proposed MIA in all experiments.
> > We attribute them mainly to the property of the methods since such trends are consistently observed across datasets.
> >
> > As discussed in our main paper, multimodal encoder methods struggle to effectively fit functions with substantial information or complexity, necessitating the amortized encoder to solve the optimization problem in a single forward operation.
> >
> > Unlike encoder-based baselines, ALFA represents an optimization-based approach with a distinct meta-learned module. This module dynamically and adaptively predicts learning rates and weight decay parameters of the learners at each optimization step. This adaptiveness of ALFA seems to aid the learner in converging towards better parameter solutions compared to the multimodal encoder baselines, especially when sufficient data is available.
> >
> > That said, the characteristics of climate data might also reinforce this performant behavior of ALFA. For instance, pressure is typically more consistent compared to temperature or precipitation; it's regarded as a reliable indicator for studying climate evolution due to its stability and insensitivity to micro-meteorological changes (Howells & Katz, 2019). Consequently, the optimization constraint is predominantly inclined towards prioritizing how well a learner fits to the data (i.e. memorization), making ALFA more suitable for modeling these signals compared to other multimodal encoder baselines.
> >
> > Howells & Katz, Long Term Trends in Atmospheric Pressure and its Variance, 2019 (https://arxiv.org/pdf/1902.01307.pdf)
> >
> >
> > >Q8:
> > While the section describes the use of a pre-trained model for surface normals, it lacks explicit detail on how this could affect the generalizability and transferability of the learned features. Clarification and potential ablation studies could be beneficial.
> > Could you elaborate on the role of the pre-trained model used for obtaining surface normals? How would the absence of this pre-trained model affect the results?
> >
> > A8: The pre-trained model was employed exclusively for dataset construction purposes, specifically to extract surface normal maps from RGB images, forming part of the joint multimodal CelebA functions dataset. This dataset aimed to showcase the effectiveness of our MIA in modeling complex joint multimodal visual functions. It's important to note that after the dataset construction, the pre-trained model was not involved in the training phase of any model, ensuring that the subsequent training and evaluations were independent of this pre-trained feature (surface normal) extraction process.

---

> ### Author Response · Authors · 2023-11-23
> **Official Response of the authors to Reviewer cLC2 [3/5]**
>
> > Q9: The manuscript does not delve into the qualitative implications of the MSE values reported. How do these numerical metrics translate into practical improvements? For instance, do lower MSEs correlate with visibly better reconstructions in real-world applications?
>
> A9: Lower MSE values indeed signify reduced error in signal reconstruction or prediction. As pointed out by the reviewer, however, these numerical improvements might not perfectly align with human perception or subjective judgment of signal quality. Nevertheless, we opt for it as our performance measure since MSE and its logarithmic derivative PSNR are commonly used metrics for evaluating the fidelity of reconstructions in implicit neural representations learning domains.
>
> >Q10: Given the critical nature of climate data, an analysis of how errors in the model's predictions could propagate into real-world applications would be valuable.
>
> A10: We believe understanding and mitigating such errors are fundamental to ensuring the reliability and practicality of any machine learning framework, and thereby we value the reviewer's insightful suggestion. Inaccurate climate predictions have far-reaching implications, potentially impacting (1) the formulation of environmental policies and disaster management strategies, (2) industries and economic planning reliant on climate forecasts, and (3) overall societal safety and preparedness.
>
> >Q11:
> The section does not sufficiently justify the choice of the AVMNIST dataset. Given its unique challenges, why was it selected over other multimodal datasets?
> It's mentioned that MTNPs fail at approximating the audio signals properly. A deeper analysis into why this failure occurs could offer valuable insights into the limitations of existing methods, thereby contextualizing the contributions of the proposed method more effectively.
>
>
> A11: The rationale behind incorporating the AVMNIST dataset in our experiments is to showcase our approach's efficacy in handling diverse joint multimodal function modeling scenarios. Unlike datasets such as synthetic, CelebA, and climate data that commonly exhibit strong axis-aligned relationships among functions, real-world scenarios often lack such alignment. For instance, audiovisual signals present distinct function domains (or coordinate systems) between image and audio signals, posing a significant challenge for models in capturing cross-modal relationships due to the absence of explicit spatiotemporal correspondence.
>
> Regarding the failure of MTNPs in approximating audio signals within the AVMNIST dataset, the MTNPs’ architecture relies on an "across-task inference" mechanism that facilitates cross-modal information exchange among axis-aligned features (see Equation 13 or Figure 2 in the corresponding paper). In AVMNIST, however, we omitted this mechanism due to the lack of explicit coordinate correspondence between images and audios, which was inevitable to train and evaluate MTNPs in AVMNIST. It’s noteworthy that cross-modal interaction can still be captured in the latent path of the first stream, as described in the Appendix D.2.
>
> To validate the above discussion further, we conducted additional experiments investigating MTNPs' dependence on axis-aligned attention mechanisms. Specifically, we compared MTNPs' performances when trained with or without such mechanisms in axis-aligned multimodal CelebA datasets. We present the results in the table below. The results indicate that MTNPs without axis-aligned attention significantly underperformed. This suggests the essential role of axis-aligned attention in MTNPs for approximating multimodal functions, indicating potential limitations in modeling heterogeneous multimodal functions that cannot use axis-aligned attention effectively.
>
> ||RGBs||||Normals||||Sketchs||||
> |--|:--:|:--:|:--:|:--:|:--:|:--:|:--:|:--:|:--:|:--:|:--:|:--:|
> |Rmin|0.00|0.25|0.50|0.75|0.00|0.25|0.50|0.75|0.00|0.25|0.50|0.75|
> |Rmax|0.25|0.50|0.75|1.00|0.25|0.50|0.75|1.00|0.25|0.50|0.75|1.00|
> |**MTNPs (Functa)**|9.871|4.807|4.105|3.644|3.983|2.552|2.339|2.221|9.680|6.568|5.395|4.819|
> |**MTNPs (Functa) w/o across-task**|13.82|6.452|5.488|4.898|4.576|2.837|2.579|2.448|12.07|7.888|6.603|6.022|
> |**MTNPs (Composers)**|9.902|4.957|4.269|3.813|4.184|2.747|2.545|2.437|9.791|6.425|5.163|4.544|
> |**MTNPs (Composers) w/o across-task**|13.76|6.421|5.474|4.897|4.635|2.888|2.639|2.513|11.89|7.411|6.031|5.403|

---

> ### Author Response · Authors · 2023-11-23
> **Official Response of the authors to Reviewer cLC2 [4/5]**
>
> >Q12:
> The manuscript mentions that the audio signals were trimmed. However, it does not explain how this preprocessing step might affect the meta-learning process.
> Could you explain how the trimming of audio signals might have affected the results and why this preprocessing was necessary?
>
> A12: In our study, we performed two preprocessing steps on the audio signals. First, we (1) decreased the sampling rate of the audio signals, followed by (2) standardizing all audio signals to a consistent length through trimming or zero-padding. Step (2) is a common practice in audio-related domains, facilitating batching during training and evaluation, which significantly reduces computational time.
>
> The decision for step (1) was specifically driven by the complexities encountered in training the multimodal baseline model, MTNPs. MTNPs exhibit quadratic complexity with respect to the support/query set size, due to the attention mechanisms directly applied to coordinate-feature pairs within these sets. This imposes excessive demands on memory and computational resources, particularly when handling audio signals, making the training process infeasible without decreasing the sampling rate.
>
> During the preprocessing of these audio signals, we confirmed that the reduction in sampling rate did not significantly simplify the problem, ensuring that the semantic integrity or quality of the original sounds remains still. Instead, it notably reduced the redundancy within the audio data, making the training of MTNPs considerably more feasible.
>
> >Q13: While the section provides a valuable ablation study to understand the impact of various modules, it is relatively shallow. For instance, it would be beneficial to understand how each of these modules contributes to reducing overfitting or improving convergence speed.
>
> A13: We appreciate the constructive feedback. In response, we conducted a more in-depth ablation study to include all possible component combinations to analyze how each of them contribute to performance. Due to time and computational limitations, we focused on the impact of each component on vanilla Composers in the synthetic dataset. The results are below.
>
> |||Ablative components||Relative Error Reduction||
> |:--:|:--:|:--:|:--:|:--:|:--:|
> |Combination|USFTs|MSFTs|FusionMLPs|Generalization|Memorization|
> |(1)|X|X|X|0|0|
> |(2)|X|X|O|38.9|54.2|
> |(3)|O|X|X|44.4|63.6|
> |(4)|O|X|O|43.1|68.0|
> |(5)|X|O|X|86.8|71.6|
> |(6)|X|O|O|86.8|76.7|
> |(7)|O|O|X|86.7|75.5|
> |(8)|O|O|O|**88.7**|**81.6**|
>
> By comparing the results (1) and (2), we find that FusionMLPs only can greatly enhance the gradients of vanilla Composers, such as by modifying the gradient direction or magnitude, or even the ones in combination with USFTs, MSFTs or both (3 vs 4, 5 vs 6, 7 vs 8). The improvement in memorization capability is more significant when vanilla Composers is further augmented with USFTs (1,2 vs 3,4), which is aligned with the study in our main paper. This indicates that USFTs is especially beneficial for capturing modality-specific patterns in the states of the learners. Unlike USFTs, we observe that MSFTs excel in utilizing cross-modal interactions among the learners’ states (3,4 vs 5,6,7,8), boosting the generalization performances of Composers significantly. Lastly, the most optimal performance is achieved when utilizing a combination of USFTs, MSFTs, and FusionMLPs, underlining the indispensable roles of each component within SFTs. This result validates the unique and crucial contribution of each component to the overall effectiveness of Composers.
>
> >Q14: The section uses relative error reduction as a metric but does not justify why this is an appropriate measure of performance. It might be valuable to consider other metrics like F1-score or ROC AUC, especially when comparing across multiple modalities.
>
> A14: We thank the suggestion. We didn't explore the use of F1-score or ROC AUC since these metrics are designed for assessing (binary) classifiers, particularly in situations where classes are imbalanced.
>
> As our paper primarily focuses on regression problems on joint multimodal continuous functions, we opted for alternative evaluation metrics, i.e. the average relative error reduction across sampling ratio ranges and modalities. The reasons are two-folded: (1) it remains robust against widely varying MSE magnitudes or scales influenced by modalities and sampling ratios, and (2) it offers a holistic understanding of the individual components' impacts within our ablation study.
>
> It's worth noting that Kim et al. also employed the same metric (referred to as relative performance gain in their paper) for similar reasons and purposes. They utilized this metric in their investigation of MNTPs' capability to capture cross-modal relationships (please refer to Section 5.3 and Table 4 in Kim et al.).
>
> Reference: Donggyun Kim, Seongwoong Cho, Wonkwang Lee, and Seunghoon Hong. Multi-task neural processes. In ICLR, 2022.

---

> > ### Author Response · Authors · 2023-11-23
> > **Official Response of the authors to Reviewer cLC2 [5/5]**
> >
> > >Q15:
> > Given the complex architecture involving USFTs, MSFTs, and Fusion MLPs, a discussion on model interpretability would be pertinent. This is essential for real-world applications where understanding model decisions is crucial.
> > Given the complexity of your model, how do you address the challenge of interpretability? Could you discuss any measures or future work planned to make the model's decisions interpretable?
> >
> > A15: Yes, the interpretability of the models is crucial. To address this concern, we conducted additional experiments to better understand the internal mechanism of how learners interact with each other and how their learning trajectories are affected through our SFTs. We are utilizing a Transformer architecture that is widely recognized and commonly used in the field. This architecture typically explains model interpretability through the utilization of attention patterns between tokens.
> > We can also infer the behavior of the model by analyzing the attention patterns of SFTs and measure the correlation with observation quality of the learners to see if SFTs adaptively exchange valuable information using the attention mechanism. More details are described in Appendix F.
> >
> > >Q16: Across all experiments, there is no discussion on how sensitive the model is to the choice of hyperparameters. This is a critical aspect to understand the robustness of the proposed methods.
> > Given the high complexity of the model, especially with the introduction of specialized modules like USFTs and MSFTs, there might be a risk of overfitting. An analysis or discussion on this would be beneficial.
> >
> > A16:
> > We acknowledge the concerns on sensitivity to hyperparameters and potential overfitting. To address this, we performed additional experiments on the Synthetic dataset, where we gradually adjusted the width and depth of the transformers in our SFTs and the multimodal encoder baseline Encoder. Notably, in our main paper, the width and depth of the transformers in SFTs and Encoder were consistently set at 192 and 1, respectively.
> >
> > For comparisons, each model was trained using five different random seeds, averaging their Mean Squared Errors (MSEs) within distinct sampling ratio ranges. Then, we calculated the average relative error reduction achieved by these methods compared to our default hyperparameter settings (width: 192, depth: 1), akin to the ablation study in our main paper. The summarized results are presented in the table below.
> >
> > The results are summarized in the table below.
> > |width / depth|64 / 1|128 / 1|192 / 1|192 / 2|192 / 3|
> > |--|:--:|:--:|:--:|:--:|:--:|
> > |Functa w/ MIA|0.20|0.10|0.00|-0.07|-0.19|
> > |Functa w/ Encoder|-0.34|-0.35|-0.31|-0.79|-0.99|
> > |Composers w/ MIA|-0.44|-0.04|0.00|-0.10|-0.18|
> > |Composers w/ Encoder|-0.53|-0.50|-0.41|-1.06|-1.36|
> >
> >
> > From the table, we observe:
> > - Overfitting was observed in Functa w/ MIA, where performance gradually degraded (0.20 -> 0.10 -> 0.00 -> -0.07 -> -0.19) as the transformers widened and deepened.
> > - Except for Functa w/ MIA, other models exhibited V-curves, indicating the default hyperparameter settings (width: 192, depth: 1) were well-suited for the synthetic dataset.
> > - Despite Functa w/ MIA showing overfitting, its worst hyperparameter setting (width: 192, depth: 3) outperformed the most performant Functa w/ Encoder (width: 192, depth: 1). Moreover, its degradation was more gradual and generous compared to Functa w/ Encoder.
> > - Similarly, in Composer-based experiments, the least favorable configuration for Composer w/ MIA (width: 64, depth: 1) performed comparably to the best-performing Composer w/ Encoder (width: 192, depth: 1). Additionally, Composer w/ Encoder exhibited significant degradation with added depth.
> >
> > In summary, although a slight overfitting was observed in Functa w/ MIA, our approach demonstrated superior performance and exhibited less sensitivity to hyperparameters compared to the robust baseline, Encoder.
> >
> > >Q17: The manuscript could benefit from a clearer exposition of experimental details to ensure reproducibility.
> > Could you provide a clear list of hyperparameters used in your experiments? This information is crucial for reproducibility and for understanding the sensitivity of your model to hyperparameter changes.
> >
> > A17: Thank you for the feedback. We updated more details on the experimental details in Appendix D. In addition, we've uploaded the code for reproducing experimental results on the Synthetic dataset to OpenReview, ensuring the reproducibility. Additionally, we are actively preparing the remaining experiment codes for public availability well ahead of the camera-ready submission deadline.

---

### Official Review · Reviewer_gNAu · 2023-11-04

**Soundness:** 2 fair
**Presentation:** 2 fair
**Contribution:** 3 good
**Rating:** 5
**Confidence:** 4

**Summary:**

This paper addresses multimodal learning of implicit neural representations by meta learning. In particular, they deal with a setting where data is scarce. The authors introduce a novel optimisation-based meta-learning framework, which they call Multimodal Iterative Adaptation (MIA). They claim MIA enables continuous the interaction among independent unimodal INR learners, and therefore the cross-modal relationships can be better captured through iterative optimization steps. In addition, they introduce a meta-learning module called state fusion transformers to aggregate states of unimodal leraners. Extensive experiments are conducted including 1D synthetic functions, real-world vision, climate, and audiovisual data.

**Strengths:**

+ It seems an interesting idea to learning implicit neural representations of multimodal data.

+ The proposed state fusion transformers to aggregate the states of the unimodal learners is also new.

+ The experimental evaluation is extensive and solid.

**Weaknesses:**

- It is a bit unclear to me why the proposed iterative way of learning could be better. Both theoretical and intuitive explanation is missing since this is the core of the proposed multimodal iterative adaptation.

- The authors indicate their multimodal iterative adaptation could better handle limited data compared to gradient based algorithms. This is not explained well either.

**Questions:**

See weaknesses.

---

> ### Author Response · Authors · 2023-11-23
> **Official Response of the authors to Reviewer gNAu [1/1]**
>
> >Q1: Why MIA can better handle limited data
>
> A1: When data of one modality is limited, the computed gradients from the learner of that modality become inevitably noisy, which slows down the convergence of the learner and triggers overfitting. Interestingly, we find that our SFTs, through attention mechanisms, can incur positive knowledge transfer from the learner whose data is sufficient to the other learners with the limited data. This positive transfer compensates for low-quality gradients, enabling enhanced guidance to the learners. Moreover, SFTs prevent negative transfer as well, refraining from updating the state representations of the learner for a specific modality when it observed sufficient data. We included the detailed discussion on this analysis in Section 5.5 (marked in blue) and Appendix F.1 of the paper.
>
> >Q2: Why do we need to apply multimodal adaptation sequentially or iteratively?
>
> A2: Since the total amount of observable data is fixed during the adaptation, any attempt to update each learner with its computed naive gradients has potential to hinder the convergence or trigger overfitting, particularly when data is limited. Therefore, our meta-learning framework involves multimodal adaptation throughout all optimization steps.
>
> To validate this claim, we conducted additional experiments, wherein we compared our MIA with the two alternative ablative approaches: applying the multimodal adaptation only at the first (MA-First) or last (MA-Last) optimization step. The experimental results on the synthetic dataset are below.
>
> ||Sine|||Gaussian|||Tanh|||ReLU|||
> |--|:--:|:--:|:--:|:--:|:--:|:--:|:--:|:--:|:--:|:--:|:--:|:--:|
> |Rmin|0.01|0.02|0.05|0.01|0.02|0.05|0.01|0.02|0.05|0.01|0.02|0.05|
> |Rmax|0.02|0.05|0.10|0.02|0.05|0.10|0.02|0.05|0.10|0.02|0.05|0.10|
> |Functa|44.26|16.07|3.319|18.81|4.388|0.953|22.61|3.667|0.586|65.29|10.79|2.157|
> |w/ MA-First|14.29|5.660|1.113|2.261|1.160|0.367|3.180|0.797|0.141|13.47|2.067|0.290|
> |w/ MA-Last|10.12|4.462|1.717|1.253|0.959|0.638|1.719|0.541 |0.267|7.063|1.254|0.305|
> |**w/ MIA**|**6.386**|**2.058**|**0.547**|**1.057**|**0.571**|**0.281**|**1.285**|**0.378**|**0.131**|**5.069**|**1.012**|**0.115**|
> |Composers|37.40|16.70|5.284|5.923|3.149|1.460|14.81|4.053|1.029|48.49|11.98|3.232|
> |w/ MA-First|16.68|7.076|2.245|2.843|1.730|0.737|5.240|1.446|0.442|22.62|3.685|0.826|
> |w/ MA-Last|14.57|9.549|7.706|1.699|1.462|1.086|2.411|1.021 |0.725|10.19|2.340|0.907|
> |**w/ MIA**|**5.564**|**1.844**|**0.627**|**0.975**|**0.528**|**0.237**|**1.257**|**0.343**|**0.128**|**4.715**|**0.943**|**0.156**|
>
> As observed, MA-First consistently exhibits poor generalization (R <= 0.05), suggesting that the subsequent independent adaptation of each learner with limited data triggers overfitting. Conversely, MA-Last's late fusion strategy appears to address overfitting when data is scarce (R <= 0.05), evidenced by its improved generalization performances. However, as a notable downside, this fusion strategy tends to disrupt previously good solutions obtained with sufficient data, resulting in poorer memorization performances (R >= 0.05).
>
> Crucially, MIA consistently outperforms these single-time adaptation methods. To delve deeper, in Figure 12 in the appendix, we also visualized the evolving interactions among learners for each optimization step by examining the attention patterns across the learners within MSFTs. In short, it reveals that learners tend to initially interact extensively with each other (high off-diagonal attention scores in the beginning) through MSFTs, followed by gradually focusing more on their individual states towards the end (high diagonal attention scores in the end). This analysis further underscores MIA's adaptability in ensuring a balanced utilization of multimodal interactions, emphasizing the necessity of such an iterative multimodal adaptation scheme.

---

### Meta-Review · Area_Chair_BT1N · 2023-12-05

**Metareview:**

A. This paper introduces a meta learning framework (Multimodal Iterative Adaptation) that proposes a way to fuse unimodal learners
B. Reviewers agree that the paper's formulation oof Shifted Feature Transducers (SFTs)  is unique and novel. The reviewers also agree the paper is well written.
C. Unfortunately the reviewers seem to agree the results are not strong enough to merit acceptance to the conference. Particularly the reviewers agree that the experiments in the paper do not match the motivation. Some of the claims in the paper do not seem to be backed up by experiments in the data (such data efficiency). Due to this reason and the general consensus by reviewers that paper could be improved in further iterations, we recommend rejection.

**Justification For Why Not Higher Score:**

Reviewers agreed that the paper did not deserve acceptance. The results of the paper need to be improved.

**Justification For Why Not Lower Score:**

N/A

---

### Decision · Program_Chairs · 2024-01-16

Reject